# GENE-M1: ADVANCING CROSS-SPECIES GENOMIC DISCOVERY VIA TAXON-SPECIFIC MIXTURE-OF-EXPERTS

## ABSTRACT

Prevailing genomic foundation models rely on a uniform architecture across all species, which ignores evolutionary divergence and results in feature interference and limited cross-species generalization. To address this, we introduce **GENE-M1**, a novel Mixture-of-Experts (MoE) framework whose architecture is strictly governed by biological taxonomy. Our method is built on three core components: (1) a hierarchical expert architecture that instantiates specialized modules for taxonomic ranks (Domain, Kingdom, Phylum, Class) to enable taxon-specific processing; (2) a dynamic router that activates expert pathways aligned with a sequence's taxonomy, ensuring hierarchical feature extraction; and (3) a progressive training strategy that transfers knowledge from higher to lower taxonomic ranks for stable optimization. We also construct **GM-DATA**, a large-scale, taxonomically-aligned benchmark comprising 294 species (spanning 5 Kingdoms, 18 Phyla, 62 Classes) that provides broad, balanced coverage across major clades and includes a held-out **GM-DATA**(eval) of 15 unseen species for rigorous cross-species evaluation. Extensive experiments on this benchmark show that **GENE-M1** significantly outperforms state-of-the-art baselines in few-shot classification and unsupervised clustering, demonstrating that explicit taxonomic alignment is key to robust and interpretable genomic representation learning. We will release our model, code and dataset soon.

## 1 INTRODUCTION

With the rapid advancement of high-throughput sequencing technologies, genomic data are being generated at an unprecedented scale across a diverse range of species. This deluge of data offers unparalleled opportunities for large-scale genomic discovery Reuter et al. (2015); Lee (2023); Ambardar et al. (2016); Hu et al. (2021). In parallel, genomic foundation models Ji et al. (2021); Zhou et al. (2024a); Dalla-Torre et al. (2025); Nguyen et al. (2023); Zhou et al. (2024b); Shao & Yan (2024) have markedly advanced the field of genomic representation learning, enabling more effective sequence modeling and supporting a variety of downstream biological applications, such as promoter and enhancer prediction, transcription-factor binding analysis, and variant-effect prioritization.

Despite these advances, a significant limitation persists: existing models (like DNABERT Ji et al. (2021), DNABERT-2 Zhou et al. (2024a), Nucleotide Transformer (NT) Dalla-Torre et al. (2025), HyenaDNA Nguyen et al. (2023), and DNABERT-S Zhou et al. (2024b)) typically rely on a single, shared network architecture to process genomic sequences from all species. This **one-size-fits-all** approach fails to account for the profound evolutionary divergence and taxon-specific characteristics inherent in genomic data in Fig. 1 (a). Consequently, these models often suffer from **feature interference** across taxa, limited cross-species generalization, and an inability to produce embeddings that align with biological taxonomy in Fig. 1 (b).

To overcome these challenges, we introduce **GENE-M1** a novel Mixture-of-Experts (MoE Cai et al. (2025)) framework explicitly aligned with biological taxonomic hierarchies (Domain → Kingdom → Phylum → Class) Shazeer et al. (2017); Ruggiero et al. (2015) in Fig. 1 (a). At its core, our approach employs a hierarchical expert architecture in which specialized modules operate at each

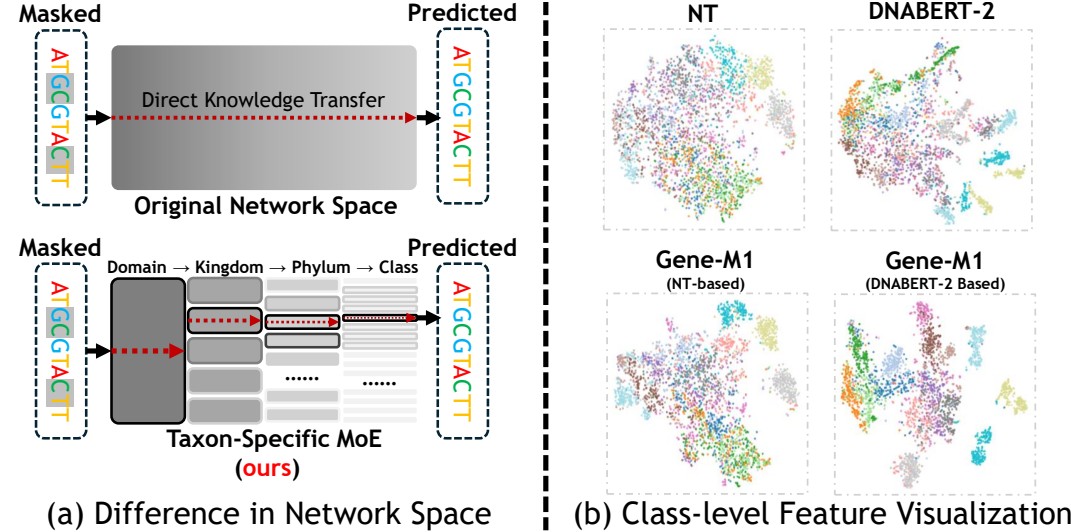

Figure 1: Motivation. (a) Baseline models process genomic data with a single shared network, lacking expert specialization and suffering from feature interference across taxa. In contrast, **GENE-M1** introduces taxon-specific experts aligned with biological hierarchy, enabling disentangled and specialized representations.(b) At the Class level, feature visualizations show that baseline models collapse diverse taxa into overlapping clusters, while **GENE-M1** produces well-separated, biologically meaningful clusters, demonstrating stronger taxon-specific customization and alignment with taxonomy.

taxonomic level, enabling taxon-specific processing (Sec. 3.1). Complementing this, a router mechanism dynamically activates experts along pathways that facilitate hierarchical feature extraction, ensuring the model structurally mirrors biological taxonomy while tailoring computation to genomic sequences from distinct taxonomic groups (Sec. 3.2). Furthermore, We employ a progressive training strategy that transfers coarse-grained knowledge from higher taxonomic levels to finer-grained levels, promoting stable optimization and hierarchical representation learning (Sec. 3.3).

To support this effort, we constructed a new, large-scale genomic dataset comprising 294 representative species from NCBI Geer et al. (2010). Unlike previous resources that are often dominated by bacteria and limited to a handful of coarse categories, our dataset, **GM-DATA**, is systematically organized according to biological taxonomy into 5 Kingdoms, 18 Phyla, and 62 Classes. This fine-grained design ensures both broad coverage across major clades (including underrepresented groups such as Archaea and Plants) and a relatively balanced representation across taxonomic ranks. To rigorously evaluate cross-species generalization, we further curate the **GM-DATA**(eval) consisting of 15 entirely unseen species, spanning 4 Kingdoms, 10 Phyla, and 15 Classes. This split enables a principled multi-level assessment of model performance under true out-of-distribution conditions. The remaining 279 species were used to construct the training set, denoted as **GM-DATA**(train), which served as the basis for model training.

Extensive experiments on supervised few-shot classification and unsupervised clustering tasks demonstrate that our taxonomy-aligned framework consistently surpasses other state-of-the-art (SOTA) DNA foundation model baselines. Moreover, our analysis indicates that different taxonomic levels provide complementary signals to representation learning, and the hierarchical design yields biologically meaningful separation of taxa. These results highlight the role of taxonomy-aware design in achieving robust and interpretable cross-species generalization.

In summary, our contributions are threefold:

- We propose **GENE-M1**, the first MoE architecture for genomics that is strictly aligned with biological taxonomy. It incorporates lightweight, hierarchical experts, dynamic routing, and a progressive training strategy to achieve improved generalization and interpretability.

- We curate and release a large-scale, taxonomically structured genomic dataset, **GM-DATA** that supports the development and evaluation of taxon-aware foundation models.

- We demonstrate through comprehensive experiments that **GENE-M1** significantly advances the state-of-the-art in cross-species genomic discovery, offering both performance gains and biological interpretability.

## 2 RELATED WORKS

**Genomic Foundation Models.** The conceptualization of DNA sequences as a language composed of nucleotides has spurred the development of various foundation models Theodoris (2024). DNABERT pioneered this direction by adapting the BERT architecture with k-mer tokenization, demonstrating the transferability of language modeling paradigms to genomics Ji et al. (2021). However, its fixed k-mer representations lacked flexibility across evolutionarily distant species Moeckel et al. (2024); Çelikkanat et al. (2024). DNABERT-2 introduced byte-pair encoding (BPE Sennrich et al. (2015)) for subword tokenization and scaled training across multiple species, improving generalization Zhou et al. (2024a). Despite this, the model retained a monolithic architecture without taxon-specific specialization. Nucleotide Transformer (NT) further scaled pre-training to over 800 species and extended context lengths, achieving strong performance but still processing all sequences through a single shared backbone, leading to potential feature interference Dalla-Torre et al. (2025). HyenaDNA replaced self-attention with long convolutions to efficiently handle million-base contexts Nguyen et al. (2023), yet it did not incorporate modularity for disentangling species-specific features. While VQDNA employed vector quantization to capture multi-resolution semantics Li et al. (2024), it overlooked explicit taxonomic alignment. DNABERT-S incorporated species-aware signals via contrastive learning to enhance clustering Zhou et al. (2024b), but its backbone remained unchanged and did not structurally reflect cross-species biological taxonomy.

In summary, existing genomic foundation models universally rely on a one-size-fits-all architecture. This design fails to account for evolutionary divergence, causing feature conflicts and limiting cross-species generalization. Our work addresses this critical gap by introducing a taxonomy-aware Mixture-of-Experts framework that explicitly aligns model structure with biological hierarchy.

**Mixture-of-Experts Architectures** Mixture-of-Experts (MoE) models have demonstrated remarkable scalability in various domains by sparsely activating specialized sub-networks, thereby achieving greater parameter efficiency Mu & Lin (2025). Seminal works such as GShard Lepikhin et al. (2020) and Switch Transformer Fedus et al. (2022) established the effectiveness of MoE for large-scale language modeling. In computational biology, AIDO.Protein Sun et al. (2024) successfully applied MoE to protein sequences, significantly improving training and inference efficiency.

In this work, we explore how MoE can be leveraged for taxonomic alignment to improve cross-species genomic modeling. Our approach overcomes the limitations of existing methods through a hierarchical routing mechanism explicitly aligned with biological taxonomy, enabling better generalization to unseen species and interpretable specialization across taxonomic levels.

## 3 METHODOLOGY

In this section, we propose **GENE-M1** a taxonomy-aligned Mixture-of-Experts framework that systematically addresses cross-species genomic modeling through three interconnected components: (1) a hierarchical expert architecture (Sec. 3.1) that processes sequences along biological taxonomy from Domain to Class levels, (2) a weighted routing mechanism (Sec. 3.2) that dynamically combines class-specific features through supervised learning Yang et al. (2015), and (3) a coarse-to-fine training strategy (Sec. 3.3) that progressively transfers knowledge while preventing feature interference. The overview of the structure is shown in Fig. 2, and this integrated approach enables **GENE-M1** to capture both universal genomic patterns and taxon-specific characteristics while maintaining parameter efficiency and biological interpretability.

### 3.1 TAXON-SPECIFIC MIXTURE-OF-EXPERTS

To address the limitations of one-size-fits-all genomic foundation models, we introduce a hierarchical Mixture-of-Experts (MoE) Ng & Deisenroth (2014) architecture that explicitly mirrors biological taxonomic organization. Our framework decomposes the representation learning process along the

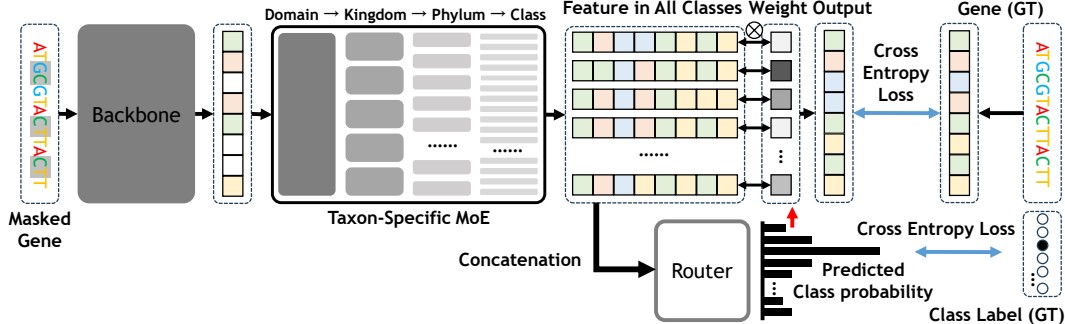

Figure 2: Overview of the training process of **GENE-M1**. The model is trained on **GM-DATA**(train), a hierarchical dataset derived from cross-species reference genomes, using a progressive freeze–unfreeze paradigm that refines knowledge from Domain → Kingdom → Phylum → Class. The objective combines masked language modeling (MLM) with router supervision to ensure taxonomy-aligned expert selection and robust cross-species generalization.

natural hierarchy of life: Domain → Kingdom → Phylum → Class, enabling specialized processing at each taxonomic level.

**Hierarchical Expert Architecture** Let $\mathcal{H} \in \mathbb{R}^{B \times L \times D}$ denote the hidden representations extracted by a DNA foundation model backbone, where $B$ is the batch size, $L$ is the sequence length, and $D$ is the hidden dimension. For each taxonomic level $\ell \in \{\text{Domain}, \text{Kingdom}, \text{Phylum}, \text{Class}\}$, we instantiate a set of expert networks $\{E_\ell^{(1)}, E_\ell^{(2)}, \ldots, E_\ell^{(N_\ell)}\}$, where $N_\ell$ corresponds to the number of taxonomic groups at level $\ell$.

Each expert module implements a transformation that first applies normalization and regularization, followed by a residual connection, and finally a dimension-reducing projection:

$$E_\ell^{(i)}(\mathbf{h}) = f_{\text{proj}}^{(\ell,i)}\left(\mathbf{h} + \texttt{Dropout}\left(\texttt{LayerNorm}(\mathbf{h})\right)\right), \tag{1}$$

where the projection function employs dimension reduction with ratio $r = N_\ell$:

$$f_{\text{proj}}^{(\ell,i)}(\mathbf{x}) = \texttt{GELU}\left(\mathbf{W}^{(\ell,i)}\mathbf{x}\right), \quad \mathbf{W}^{(\ell,i)} \in \mathbb{R}^{D \times D/r}. \tag{2}$$

This design ensures parameter efficiency while maintaining cross-species representational capacity.

**Taxonomy-Aligned Composition** The hierarchical composition processes sequences through all experts at each taxonomic level, with feature concatenation enabling comprehensive taxonomic representation. The transformation is defined as:

$$\mathbf{Y} = E_{\text{Class}}^{(k)} \circ \bigoplus_{j=1}^{N_{\text{Phylum}}} E_{\text{Phylum}}^{(j)} \circ \bigoplus_{i=1}^{N_{\text{Kingdom}}} E_{\text{Kingdom}}^{(i)} \circ E_{\text{Domain}}(\mathbf{X}) \tag{3}$$

where $\mathbf{X} \in \mathbb{R}^{B \times L \times D}$ is the input representation, $\mathbf{Y} \in \mathbb{R}^{B \times L \times \frac{D}{N_{\text{Class}}} \times N_{\text{Class}}}$ is the output representation, $\circ$ denotes function composition, and $\bigoplus$ represents concatenation along the feature dimension. This composition strategy progressively incorporates taxonomic information, with each level adding specialized representations that are concatenated to form a comprehensive feature set.

## 3.2 WEIGHTED ROUTING FOR CLASS-SPECIFIC FEATURES

The routing mechanism dynamically weights expert contributions based on taxonomic characteristics, enabling adaptive feature combination that aligns with the biological hierarchy. This approach ensures that the model can specialize its processing based on the taxonomic properties of each input sequence.

**Context-Aware Routing** Given the concatenated representations from all class experts, we compute routing weights using a linear classifier that learns to identify the most relevant taxonomic features:

$$\mathbf{s} = \mathbf{W}_{\text{router}} \cdot \texttt{Flatten}(\mathbf{Y}), \tag{4}$$

where $\mathbf{W}_{\text{router}} \in \mathbb{R}^{N_{\text{Class}} \times (B \cdot L \cdot D)}$ contains the learnable routing parameters, and `Flatten` reshapes the tensor from 3D to 2D by collapsing the batch and sequence length dimensions. The routing weights are normalized using softmax with temperature scaling to produce a probability distribution over class experts:

$$\alpha_i = \frac{\exp(s_i/\tau)}{\sum_{j=1}^{N_{\text{Class}}} \exp(s_j/\tau)}, \tag{5}$$

where $\tau > 0$ is a temperature hyperparameter that controls the sharpness of the weight distribution. Lower values of $\tau$ produce more peaked distributions, favoring specialization, while higher values encourage more uniform expert utilization.

Since taxonomic labels are available during training, we supervise the router using cross-entropy loss to ensure that expert selection aligns with biological taxonomy:

$$\mathcal{L}_{\text{router}} = -\frac{1}{B} \sum_{b=1}^{B} \sum_{i=1}^{N_{\text{Class}}} y_{b,i} \log \left( \frac{\exp(s_{b,i})}{\sum_{j=1}^{N_{\text{Class}}} \exp(s_{b,j})} \right), \tag{6}$$

where $y_{b,i}$ is the one-hot encoded taxonomic label for sequence $b$ at the class level. This supervision encourages the router to assign higher weights to experts corresponding to the correct taxonomic classification.

**Weighted Feature Concatenation**    The final representation is obtained by weighting and concatenating class-specific features according to the learned routing weights:

$$\mathbf{Z} = \bigoplus_{i=1}^{N_{\text{Class}}} \alpha_i \cdot \mathbf{Y}^{(i)}, \tag{7}$$

where $\mathbf{Y}^{(i)}$ denotes the portion of $\mathbf{Y}$ corresponding to class expert $i$ (with dimensionality $D/N_{\text{Class}}$), and $\mathbf{Z} \in \mathbb{R}^{B \times L \times D}$ is the final routed representation. This weighted concatenation ensures that the model can adaptively combine taxonomic features based on the input sequence, with experts receiving higher weights for sequences that match their taxonomic specialization. The resulting representation $\mathbf{Z}$ captures both universal genomic patterns and taxon-specific characteristics, providing a rich feature set for downstream tasks.

### 3.3 Taxon Coarse-to-Fine Learning

We employ a progressive training strategy that mimics evolutionary specialization, transferring knowledge from coarse to fine taxonomic levels Fidler et al. (2010).

**Sequential Training with Layer Freezing**    The training proceeds sequentially through four phases, where at each phase $t$ we optimize parameters $\theta_t$ while keeping all previously trained parameters $\theta_{<t}$ frozen:

$$\theta_l^* = \arg\min_{\theta_t} \mathcal{L}(\theta_t | \theta_{<t} \text{ frozen}), \quad t = 1, \ldots, 4 \tag{8}$$

The sequence follows the taxonomic hierarchy: $(t = 1)$ Domain $\rightarrow$ $(t = 2)$ Kingdom $\rightarrow$ $(t = 3)$ Phylum $\rightarrow$ $(t = 4)$ Class, with each level initialized from its parent and trained on increasingly specialized data subsets. This sequential freezing strategy ensures stable optimization by preventing feature interference between taxonomic levels, as each level specializes based on fixed representations from coarser levels.

**Multi-Objective Optimization**    The training objective combines masked language modeling with router supervision:

$$\mathcal{L}_{\text{total}} = \lambda_{\text{mlm}} \mathcal{L}_{\text{mlm}} + \lambda_{\text{router}} \mathcal{L}_{\text{router}}, \tag{9}$$

where $\mathcal{L}_{\text{mlm}}$ is the standard masked language modeling loss and $\mathcal{L}_{\text{router}}$ is the routing supervision loss defined in Sec. 3.2. The hyperparameters $\lambda_{\text{mlm}}$ and $\lambda_{\text{router}}$ balance the two objectives.

This coarse-to-fine learning approach ensures that the model progressively incorporates taxonomic specialization while maintaining the stability of previously learned representations, effectively addressing feature interference and enabling robust cross-species generalization.

## 4 EXPERIMENTS

In this section, we present a comprehensive experimental evaluation of **GENE-M1**. We first construct a large-scale, taxonomically structured dataset (**GM-DATA**) and establish principled training and test partitions to ensure strict cross-species evaluation. We then describe training configurations and experimental settings, including backbone models and task definitions. Finally, we assess the framework through three complementary perspectives—cross-species gene classification, cross-species gene clustering, and experimental analysis—providing a systematic evaluation of predictive accuracy, representation quality, and the role of hierarchical modeling.

### 4.1 DATA CONSTRUCTION

**Data Collection and Composition** We constructed a comprehensive genomic dataset systematically collected from the NCBI RefSeq database Geer et al. (2010). The dataset comprises 294 representative species, organized strictly according to biological taxonomy into 5 kingdoms, 18 phyla, and 62 classes. All genomes were preprocessed following a standardized pipeline, After filtering, we further replace non-ATCG characters with N. The resulting sequences are segmented into 6,000 base-pair fragments with 100 base-pair overlaps, merged into a list of chunks, and finally converted into a HuggingFace dataset.Together, these steps yield a high-quality, taxonomically structured dataset (**GM-DATA**), that supports robust training and evaluation of our framework.

**Training and Test Partitioning** To establish a principled evaluation protocol, we partition the dataset into a training set and a held-out evaluation set. Specifically, 279 species are assigned to the training set (**GM-DATA**(train) see Appendix C.2), while 15 phylogenetically diverse and previously unseen species are reserved for the test set (**GM-DATA**(eval) see Appendix C.3). This split ensures that evaluation is performed under strict cross-species conditions, thereby probing the model's ability to generalize beyond seen taxa. The full taxonomic organization, spanning from Domain to Class, is preserved in both splits, providing a systematic framework for multi-level assessment.

**Taxonomic Coverage Comparison** As detailed in Table 1, our dataset provides superior taxonomic balance compared to existing benchmarks. Compared with previous datasets that are heavily skewed toward bacteria

Table 1: Comparison of genomic datasets.

| Dataset | Total Species | Training/Test Split | Taxonomic Levels |
|---|---|---|---|
| DNABERT-2 | 135 | 135/0 | 7 coarse groups |
| NT | 850 | 850/0 | 6 coarse groups |
| **Ours** | 294 | 279/15 | 5 Kingdoms, 18 Phyla, 62 Classes |

(74–78%), our collection provides a more balanced distribution across kingdoms (Fungi: 13%, Animalia: 22%, Plantae: 16%, Bacteria: 34%, Archaea: 15%), thereby reducing bias toward any single lineage and ensuring cross-kingdom balance. Moreover, our dataset preserves the full biological hierarchy (Domain $\rightarrow$ Kingdom $\rightarrow$ Phylum $\rightarrow$ Class; see Appendix C.1), which not only supports learning at multiple granularities but also enhances biological interpretability by aligning representations with natural taxonomy.

### 4.2 SETTINGS

**Training Configuration** For optimization, we employ the AdamW optimizer Loshchilov & Hutter (2017) with an initial learning rate of $1 \times 10^{-3}$, combined with a cosine learning rate scheduler and a 10% warm-up phase. The effective batch size is set to 64 via gradient accumulation. Gradient clipping (maximum norm = 1.0) and mixed-precision training are applied to enhance stability and efficiency. The overall training objective is empirically formulated to jointly optimize masked language modeling (MLM) loss and router supervision loss, with weights $\lambda_{\mathrm{mlm}} = 1.0$ and $\lambda_{\mathrm{router}} = 0.2$, respectively.

**Backbone & Tasks** We instantiate **GENE-M1** based on two representative genomic foundation models, NT Dalla-Torre et al. (2025) and DNABERT-2 Zhou et al. (2024a). To evaluate the proposed framework, we design two complementary protocols: **Cross-Species Gene Classification** across unseen species; **Cross-Species Gene Clustering** for alignment of learned embeddings.

### 4.3 EVALUATION ON CROSS-SPECIES GENE CLASSIFICATION

Table 2: Model performance (macro-F1, ↑) at different taxonomic levels and shot numbers. The best results are highlighted in red, and the second best in blue. ↑ indicates that higher F1 scores are better.

| Level | Shots | Models | | | | | |
|---|---|---|---|---|---|---|---|
| | | NT | DNA BERT-2 | DNA BERT-S | HyenaDNA | **GENE-M1** (NT Based) | **GENE-M1** (DNABERT-2 Based) |
| Kingdom | 1-shot | 47.67% | 46.08% | 51.16% | 19.25% | 39.79% | 77.42% |
| | 2-shot | 44.59% | 54.22% | 52.70% | 24.37% | 43.12% | 75.14% |
| | 5-shot | 39.86% | 68.06% | 74.04% | 24.72% | 51.05% | 85.36% |
| | 10-shot | 57.55% | 72.82% | 79.49% | 32.83% | 61.34% | 87.52% |
| | 20-shot | 66.75% | 79.30% | 83.06% | 36.53% | 78.15% | 91.20% |
| Phylum | 1-shot | 24.79% | 22.09% | 33.70% | 8.44% | 60.04% | 58.20% |
| | 2-shot | 23.59% | 35.60% | 45.69% | 11.15% | 51.91% | 62.40% |
| | 5-shot | 30.80% | 42.79% | 51.05% | 13.79% | 53.76% | 74.33% |
| | 10-shot | 36.88% | 45.28% | 60.54% | 17.76% | 56.93% | 75.41% |
| | 20-shot | 50.96% | 53.19% | 63.60% | 21.90% | 79.75% | 80.34% |
| Class | 1-shot | 14.48% | 15.82% | 26.45% | 4.12% | 23.21% | 47.45% |
| | 2-shot | 10.91% | 29.29% | 37.04% | 5.89% | 31.90% | 60.56% |
| | 5-shot | 18.46% | 34.34% | 48.05% | 9.21% | 48.28% | 71.73% |
| | 10-shot | 26.22% | 41.27% | 50.62% | 13.01% | 54.87% | 73.26% |
| | 20-shot | 36.34% | 44.58% | 55.91% | 14.69% | 65.47% | 78.84% |

We evaluate downstream performance on the **GM-DATA**(eval) by constructing three hierarchical classification tasks at the kingdom, phylum, and class levels, systematically assessing the model's cross-species classification capability at varying taxonomic granularities. Performance is measured using the macro-averaged F1 score, which balances precision and recall across classes and is particularly suited for imbalanced taxonomic distributions Goutte & Gaussier (2005).In addition, we adopt few-shot training regimes (1-shot, 2-shot, 5-shot, 10-shot, and 20-shot), where models are trained with only a handful of labeled samples and subsequently evaluated. Such few-shot evaluations are particularly valuable, as they provide a rigorous assessment of the base model's adaptability and robustness in cross-species scenarios Wang et al. (2020).

**Results** As shown in Table 2, **GENE-M1** achieves state-of-the-art (SOTA) performance across all few-shot training settings on the hierarchical classification tasks. Notably, at the class level, the macro-averaged F1 score exceeds that of DNABERT-2 Zhou et al. (2024a) by more than 25% under every few-shot condition, highlighting the advantage of taxonomy-aligned modeling for fine-grained cross-species classification.

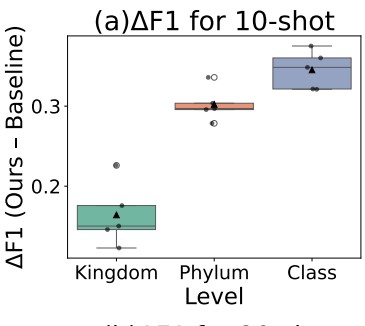

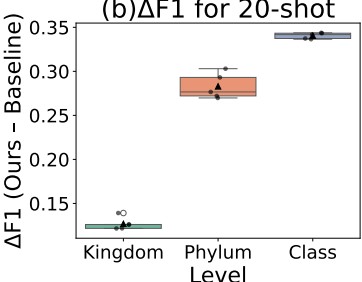

Figure 3: Box plot of ΔF1 improvements across taxonomy levels under 10-shot and 20-shot setting.

**Analysis** The most pronounced and consistent improvements are observed at the class level, where fine-grained discrimination is inherently more difficult. Furthermore, this trend is corroborated by the boxplot analysis in Fig. 3, where ΔF1 denotes the performance gain of **GENE-M1** over baseline models. The results reveal substantially larger ΔF1 margins at the class level compared to coarser taxonomic ranks.Since the evaluation is performed on the **GM-DATA**(eval) which contains only previously unseen species, the observed performance improvements of **GENE-M1** provide direct evidence of its enhanced cross-species generalization capability.

Table 3: Model performance across taxonomic levels with clustering evaluation metrics (↑ indicates higher is better). Best results are highlighted in red, and second best in blue.

| Level | Metric↑ | NT | DNA BERT-2 | DNA BERT-S | HyenaDNA | GENE-M1 (NT Based) | GENE-M1 (DNABERT-2 Based) |
|---|---|---|---|---|---|---|---|
| Kingdom | ACC | 44.63% | 40.38% | 54.38% | 41.63% | 44.38% | 60.25% |
| | NMI | 15.05% | 9.19% | 26.27% | 14.83% | 18.41% | 45.48% |
| | ARI | 12.63% | 7.44% | 18.48% | 11.29% | 13.39% | 31.41% |
| Phylum | ACC | 42.85% | 30.50% | 38.30% | 26.60% | 47.80% | 56.55% |
| | NMI | 37.38% | 21.47% | 31.24% | 23.65% | 43.72% | 57.98% |
| | ARI | 24.10% | 11.56% | 18.22% | 11.46% | 28.22% | 40.54% |
| Class | ACC | 34.20% | 23.53% | 29.87% | 19.83% | 37.03% | 44.83% |
| | NMI | 33.09% | 23.64% | 27.52% | 17.96% | 39.84% | 56.20% |
| | ARI | 17.48% | 10.02% | 13.37% | 7.16% | 22.09% | 32.66% |

## 4.4 Evaluation on Cross-Species Gene Clustering

We perform unsupervised cross-species clustering on **GM-DATA**(eval) to evaluate whether models can recover taxonomic structures from embedding similarity alone. DNA embeddings from baselines and **GENE-M1** are projected into a shared latent space, followed by $k$-means He & Li (2024) to test if unseen species form compact intra-taxon clusters with clear inter-taxon boundaries.

**Results** As shown in Table 3, we report results using three standard clustering metrics: Clustering Accuracy (ACC), Normalized Mutual Information (NMI), and Adjusted Rand Index (ARI) (see Appendix B for detailed definitions). **GENE-M1** achieves SOTA performance across all evaluation metrics and taxonomic levels. The most pronounced improvements are observed at the class level. This suggests that other models tend to produce overlapping clusters with blurred boundaries at fine-grained levels, whereas **GENE-M1** effectively mitigates feature interference, leading to more compact intra-class distributions and the establishment of clearer and more distinguishable inter-class boundaries, this conclusion is further supported by visualization, as shown in Fig. 4.

**Analysis** These findings provide strong evidence that **GENE-M1** enhances cross-species generalization by disentangling lineage-specific signals within a unified hierarchical framework. By progressively refining representations from coarse (kingdom) to fine-grained (class) levels, the model not only reduces ambiguity in distinguishing closely related taxa but also yields embeddings that remain biologically meaningful for unseen species.

## 4.5 Ablation Study & Analysis

**Cross-Species Feature Decoupling** We conduct an ablation analysis to assess how hierarchical depth influences representation quality. As shown in Fig. 5, embeddings evolve progressively from coarse to fine: Domain-level representations remain entangled, while Kingdom and Phylum layers exhibit clearer boundaries, and the Class layer achieves the most distinct separation.

Quantitative clustering results in Table 4 confirm this trend. ACC, NMI, and ARI consistently improve as depth increases. These findings indicate that the hierarchical expert modules, together with the router, progressively refine genomic representations, yielding superior fine-grained discrimination across taxa. By disentangling lineage-specific signals at multiple hierarchical levels, the model enhances its ability to generalize to previously unseen species, thereby strengthening cross-species adaptability.

Table 4: Performance across taxonomic levels with clustering evaluation metrics. Colors indicate relative magnitude, from low to high: orange → teal → blue → red.

| Level | Metric↑ | Layers | | | |
|---|---|---|---|---|---|
| | | Domain | Kingdom | Phylum | Class |
| Kingdom | ACC | 58.13% | 60.00% | 59.50% | 60.25% |
| | NMI | 26.87% | 36.62% | 41.58% | 45.48% |
| | ARI | 19.69% | 31.98% | 30.16% | 31.34% |
| Phylum | ACC | 36.35% | 41.15% | 46.85% | 56.55% |
| | NMI | 30.17% | 44.32% | 50.16% | 57.98% |
| | ARI | 16.37% | 26.21% | 31.55% | 40.54% |
| Class | ACC | 28.07% | 35.80% | 44.03% | 44.83% |
| | NMI | 25.90% | 43.38% | 50.84% | 56.20% |
| | ARI | 11.84% | 21.75% | 30.36% | 32.66% |

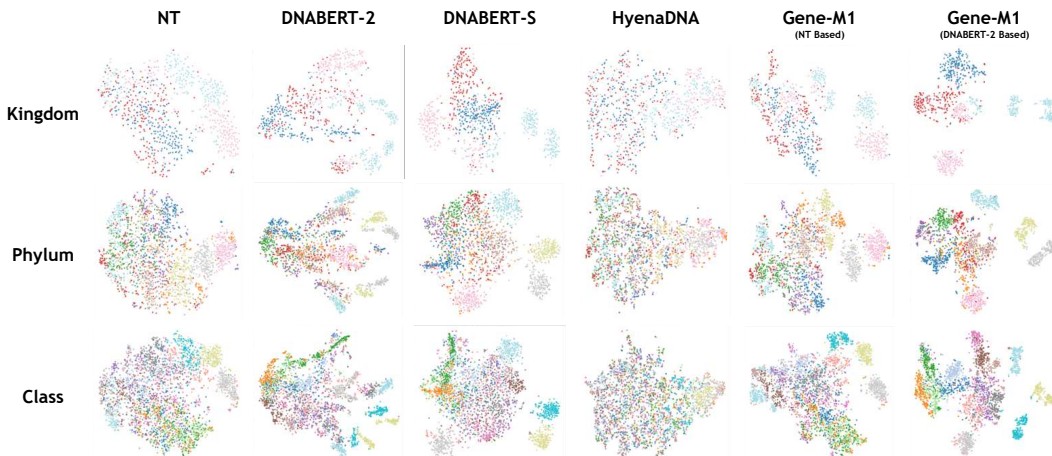

Figure 4: T-SNE visualizations of sequence embeddings across Kingdom, Phylum, and Class. Compared with baselines, **GENE-M1** produces clearer, more separable clusters aligned with taxonomy.

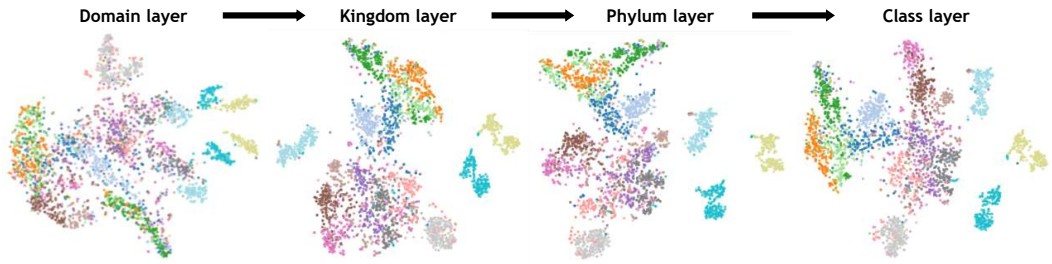

Figure 5: T-SNE visualizations of embeddings from different hierarchical levels (Domain, Kingdom, Phylum, Class) in **GENE-M1** (**DNABERT-2 Based**), showing progressive refinement from coarse to fine-grained taxonomy.

To further investigate how the router allocates experts, we visualize the routing weights in the form of a heatmap, where each row corresponds to a species and each column denotes an expert. The color intensity reflects the routing weight assigned to that expert.

**Weighted Routing** As shown in Fig. 6, different species exhibit distinct activation pathways across experts, demonstrating that the router adaptively allocates taxon-specific

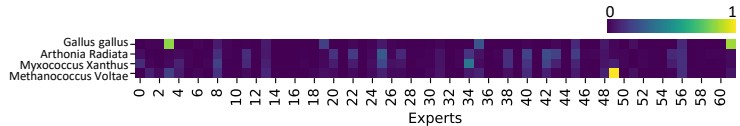

Figure 6: Router weights heatmap.

channels rather than distributing weights uniformly. The emergence of clear vertical bands further indicates that different experts consistently capture specialized biological signals, confirming that the routing mechanism effectively disentangles species-level features. These results highlight that weighted routing fosters stable specialization, reduces interference across experts, and ultimately enhances fine-grained classification performance.

## 5 CONCLUSION

In summary, we present **GENE-M1**, a taxon-specific Mixture-of-Experts framework for genomics. By structurally mirroring biological hierarchy, our model disentangles taxon-specific features and achieves superior cross-species generalization. Extensive experiments confirm that **GENE-M1** not only outperforms state-of-the-art baselines but also yields biologically meaningful and interpretable representations, thereby facilitating more effective analysis of cross-species genomic data.

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

APPENDIX ORGANIZATION

This appendix provides supplementary materials to support the main body of the paper. Section A.1 reports additional baseline training analyses, demonstrating the necessity of training all baseline models on GM-DATA to ensure fair and architecture-only comparison. Section A.2 provides extended analysis of model scale, computational cost, and their relationship to downstream performance. Section A.3 presents additional metagenomic-style clustering results, evaluating the robustness of GENE-M1 under noisy or weakly labeled taxonomic settings. Section A.4 analyzes the progressive training strategy and expert specialization, including visualizations that reveal expert collapse in non-progressive variants. Together, these supplementary experiments offer a more comprehensive understanding of how the architecture behaves under different training regimes, data conditions, and model configurations.Section A.5 provides a quantitative analysis of expert usage to assess whether GENE-M1 exhibits expert collapse. In Section B, we first define the clustering evaluation metrics used in our experiments, including ACC, NMI, and ARI, which together provide a comprehensive evaluation from accuracy, information-theoretic consistency, and clustering quality perspectives. Then, Section C.1 presents the full taxonomic hierarchy (Table 9) that forms the biological foundation of our taxonomy-aligned Mixture-of-Experts framework. Section C.2 introduces **GM-DATA**(train) used for pre-training (Table 10), while Section C.3 describes the independently constructed benchmark **GM-DATA**(eval) (Table 11) for rigorous evaluation of cross-species generalization. Beyond data resources, Section D discusses the limitations of our current dataset and modeling approach, and Section E outlines several promising directions for future work. Finally, Section F clarifies the usage of large language models (LLMs) in this work, noting that they were only employed for language polishing and had no role in model design, data construction, or experimental analysis. Together, these supplementary materials comprehensively cover evaluation metrics, data resources, current limitations, future opportunities, and LLM usage, providing a broad perspective for understanding and extending this study.

## A  EXPERIMENTS

### A.1  ADDITIONAL BASELINE TRAINING ANALYSIS

To ensure a fair and architecture-only comparison, all baseline models (NT, DNABERT-2, DNABERT-S, and HyenaDNA) were trained on the **GM-DATA**(train) split before evaluation. This setup guarantees that performance differences reflect architectural characteristics rather than artifacts arising from dataset imbalance or discrepancies in pretraining distributions.

To further assess the necessity of training on **GM-DATA**(train) we additionally compared (i) off-the-shelf baselines without any training on **GM-DATA**(train) and (ii) baselines trained on **GM-DATA**(train)Ás shown in Table 5, off-the-shelf models exhibit substantially degraded performance across all metrics, confirming that training on **GM-DATA**(train) is essential for a fair comparison and that the improvements of **GENE-M1** derive from its hierarchical MoE design rather than simple exposure to a balanced dataset.

Table 5: Comparison of off-the-shelf and trained baselines on GM-DATA (5-shot setting).

| Model | Trained on GM-DATA | Macro-F1 | ACC | NMI | ARI |
|---|---|---|---|---|---|
| NT | ✗ | 15.49% | 30.78% | 32.46% | 15.16% |
| DNABERT-2 | ✗ | 30.08% | 19.55% | 20.45% | 8.73% |
| DNABERT-S | ✗ | 44.21% | 27.34% | 26.76% | 11.54% |
| HyenaDNA | ✗ | 5.58% | 12.95% | 16.87% | 5.87% |
| NT | ✓ | 18.46% | 34.20% | 34.09% | 17.48% |
| DNABERT-2 | ✓ | 34.34% | 23.53% | 23.64% | 10.02% |
| DNABERT-S | ✓ | 48.05% | 29.87% | 27.52% | 13.37% |
| HyenaDNA | ✓ | 9.21% | 19.83% | 17.96% | 7.16% |
| GENE-M1 (NT-based) | ✓ | 48.28% | 37.03% | 39.84% | 22.09% |
| GENE-M1 (DNABERT-2–based) | ✓ | **71.73%** | **44.83%** | **56.20%** | **32.66%** |

## A.2 ADDITIONAL ANALYSIS OF MODEL SCALE AND COMPUTATIONAL COST

Although **GENE-M1** adopts a Mixture-of-Experts (MoE) design and therefore contains more total parameters and FLOPs than single-tower baselines, the performance improvement cannot be attributed merely to scaling up the model. Table 6 summarizes the parameter counts, FLOPs, training-token budgets, and downstream performance for all baselines and GENE-M1 variants.

Table 6: Model scale, computational cost, and downstream performance comparison.

| Model | Params | FLOPs | Training Tokens | Macro-F1 | ACC | NMI | ARI |
|---|---|---|---|---|---|---|---|
| NT | 480M | 3.19 | 75B | 18.46% | 34.20% | 34.09% | 17.48% |
| DNABERT-2 | 117M | 1.00 | 274B | 34.34% | 23.53% | 23.64% | 10.02% |
| DNABERT-S | 117M | 1.02 | 312B | 48.05% | 29.87% | 27.52% | 13.37% |
| GENE-M1 (NT-based) | 947M | 4.29 | 50B | 48.28% | 37.03% | 39.84% | 22.09% |
| GENE-M1 (DNABERT-2–based) | 363M | 1.72 | 25B | **71.73%** | **44.83%** | **56.20%** | **32.66%** |

The empirical evidence supports four observations:

- The additional parameters in **GENE-M1** arise from biologically structured functional decomposition, rather than generic capacity expansion.
- Larger or equally sized single-tower baselines do not outperform **GENE-M1** despite having similar or higher FLOPs.
- All models are trained on a relatively small corpus (GM-DATA (train)), limiting the potential benefits of naive scaling.
- The expert modules in **GENE-M1** can be explicitly size-controlled: deeper taxonomic experts are assigned progressively smaller hidden dimensions, enabling bounded and scalable parameter growth.

Overall, although **GENE-M1** introduces additional parameters and computational cost due to its fully activated MoE architecture, the performance gains consistently reflect the hierarchical, taxonomy-aligned expert design rather than model size. The improvements stem from structured expert specialization and biologically meaningful feature partitioning, leading to substantially stronger cross-species generalization than equally sized or larger single-tower baselines.

## A.3 ADDITIONAL ANALYSIS ON METAGENOMIC-STYLE CLUSTERING

The current formulation of **GENE-M1** relies on reasonably accurate hierarchical taxonomic labels: the router loss is supervised using explicit Kingdom/Phylum/Class assignments, and the model is primarily designed for curated reference genomes where such labels are relatively stable. For unknown or taxonomically contentious species, this assumption limits the model's applicability. We make this scope restriction explicit in the Limitation section of the revised manuscript.

To partially assess the behavior of GENE-M1 beyond clean reference genomes, we further evaluate it under a metagenomic-style clustering setting. In this setting, sequences are drawn from mixed or environmentally derived samples, where taxonomic structure is noisy or only weakly defined. Table 7 reports performance across eight subsets spanning plant and marine datasets.

Table 7: Metagenomic-style clustering accuracy across plant and marine subsets.

| Model | Plant | | | | Marine | | | | Ave |
|---|---|---|---|---|---|---|---|---|---|
| | 0 | 1 | 2 | 3 | 0 | 1 | 2 | 3 | |
| NT | 11.03% | 12.81% | 12.46% | 11.74% | 13.92% | 13.67% | 13.48% | 13.44% | 12.82% |
| DNABERT-2 | 16.25% | 15.01% | 14.83% | 17.02% | 17.76% | 18.02% | 18.44% | 17.94% | 16.91% |
| HyenaDNA | 7.96% | 8.87% | 8.16% | 8.02% | 7.19% | 7.76% | 7.15% | 6.94% | 7.76% |
| GENE-M1 (NT-based) | 16.54% | 17.32% | 17.89% | 17.19% | 19.16% | 19.72% | 19.22% | 19.73% | **18.35%** |
| GENE-M1 (DNABERT-2–based) | 20.39% | 19.45% | 20.41% | 22.15% | 25.19% | 22.31% | 21.67% | 22.71% | **21.79%** |

Introducing additional supervised router heads at the kingdom and phylum levels is a possible extension, but doing so requires maintaining multiple routers and auxiliary classification heads, increasing the number of routing components and parameters. In this work, supervision is intentionally restricted to the class level in order to control model size and isolate the contribution of the hierarchical MoE architecture itself. Exploring more flexible training schedules and multi-level router supervision, while carefully managing routing complexity, presents an interesting avenue for future work.

Across both plant and marine datasets, GENE-M1 demonstrates consistent but moderate improvements over NT, DNABERT-2, and HyenaDNA. These results indicate that the taxonomy-aligned architecture still provides benefit even when the underlying taxonomic structure is noisy. However, the improvements remain limited, confirming that GENE-M1 does not fully resolve the challenges posed by unknown or contentious species. Addressing such cases represents an important direction for future work.

### A.4    ANALYSIS OF PROGRESSIVE TRAINING AND EXPERT SPECIALIZATION

The progressive training strategy adopted in GENE-M1 imposes a structural constraint: higher-level modules are frozen during later stages, which limits the extent to which general representations can be further refined while learning finer-grained ones. This trade-off is intentional in our setting. Because supervision is provided only at the class level, the model relies on coarse-to-fine progressive training, combined with selective data routing by taxonomic level, to promote stable and hierarchy-consistent specialization of experts.

Prior large-scale MoE systems Shazeer et al. (2017); Fedus et al. (2022) have shown that jointly training all experts and the router in a single stage increases the likelihood of *expert collapse*, where most inputs are routed to only a minority of experts. In contrast, the progressive schedule helps distribute specialization across taxonomic levels.

To examine this effect, we compared the standard progressive training scheme with a non-progressive variant in which all experts are trained jointly from the beginning. As shown in Table 8, the non-progressive model yields substantially lower performance across all taxonomic levels.

Table 8: Comparison of progressive and non-progressive training, evaluated at the Kingdom, Phylum, and Class levels.

| Level | DNABERT-2 | GENE-M1 (NT-based) | Non-Progressive Training (DNABERT-2–based) |
|---|---|---|---|
| Kingdom | 7.44% | **31.41%** | 12.91% |
| Phylum | 11.56% | **40.54%** | 15.48% |
| Class | 10.02% | **32.66%** | 13.34% |

To further illustrate this collapse phenomenon, we visualize the routing distribution of the most frequently activated experts in the non-progressive setting. Specifically, we compute the average routing probability of each expert across all samples, select the top four experts with the highest activation frequency, and plot their sample-wise routing weights. As shown in Fig. 7, the majority of inputs are dominated by only one or two experts, while the remaining experts receive substantially lower activation. This pattern provides direct evidence of expert collapse and highlights the necessity of the progressive coarse-to-fine training strategy for achieving stable and semantically meaningful expert specialization in GENE-M1.

### A.5    QUANTITATIVE ANALYSIS OF EXPERT USAGE AND ROUTING ENTROPY

To complement the qualitative visualizations of routing behavior, this section provides a quantitative analysis of expert usage to assess whether GENE-M1 exhibits expert collapse. In our implementation, routing is instantiated at the class level with 62 experts, and the MoE architecture is dense rather than top-$k$ sparse. To characterize load balance, we compute routing entropy at both the token level and the global expert-load level.

On the test set, the normalized token-level routing entropy is 0.85 and the normalized global entropy is 0.91, where a value of 1.0 corresponds to perfectly uniform expert utilization. These results

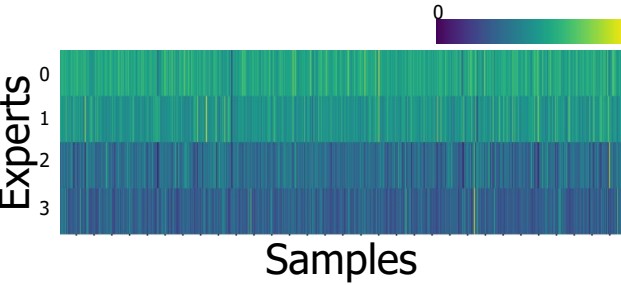

Figure 7: Routing heatmap of the top four most frequently activated experts in the non-progressive variant. Most inputs are routed predominantly through one or two experts, indicating a strong collapse effect.

indicate that routing mass is distributed across a large number of experts rather than concentrated on a small subset. This behavior aligns with the qualitative heatmaps presented earlier and suggests that GENE-M1 does not exhibit severe expert collapse in practice. The entropy-based statistics provided here offer a complementary quantitative perspective on expert load balancing.

## B  EVALUATION METRICS

In this section, we provide definitions of the clustering evaluation metrics used in our experiments.

**Accuracy (ACC).**  ACC measures the proportion of correctly clustered samples after an optimal permutation of cluster labels:

$$\text{ACC} = \frac{1}{n} \max_{m \in \mathcal{M}} \sum_{i=1}^{n} \mathbf{1}\{y_i = m(c_i)\},$$

where $y_i$ and $c_i$ denote the ground-truth and predicted labels, and $m$ is a one-to-one mapping.

**Normalized Mutual Information (NMI).**  NMI quantifies the mutual dependence between ground-truth labels $Y$ and predicted clusters $C$, normalized by their entropies:

$$\text{NMI}(Y, C) = \frac{2 \cdot I(Y;C)}{H(Y) + H(C)},$$

where $I(\cdot; \cdot)$ is mutual information and $H(\cdot)$ is entropy.

**Adjusted Rand Index (ARI).**  ARI evaluates the similarity between two partitions by counting pairwise agreements, adjusted for chance:

$$\text{ARI}(Y, C) = \frac{\text{RI}(Y, C) - \mathbb{E}[\text{RI}(Y, C)]}{\max(\text{RI}(Y, C)) - \mathbb{E}[\text{RI}(Y, C)]},$$

where RI is the Rand Index.

These three metrics together provide a comprehensive evaluation: ACC reflects classification accuracy, NMI captures information-theoretic consistency, and ARI measures clustering quality while accounting for random labeling.

## C  DATA

### C.1  TAXONOMIC HIERARCHY

Table 9 presents the complete taxonomic hierarchy used in this study. The structure follows the principles of biological systematics, where Domains encompass all Kingdoms, each of which is further divided into Phyla and Classes. This hierarchical organization provides the foundational framework for our taxonomy-aligned Mixture-of-Experts model, ensuring that representation learning remains consistent with biological taxonomy across different levels of granularity.

Table 9: Taxonomic hierarchy used in our model, where each Domain encompasses multiple Kingdoms (Domain → Kingdom → Phylum → Class).

| Kingdom | Phylum | Class |
|---|---|---|
| **Animalia** | Chordata | Amphibia, Actinopterygii, Reptilia, Aves, Mammalia |
| | Arthropoda | Insecta, Arachnida, Crustacea |
| | Mollusca | Gastropoda, Bivalvia |
| **Plantae** | Angiosperms | Liliopsida, Magnoliopsida |
| | Bryophyta | Bryopsida, Marchantiopsida |
| | Algae | Chlorophyta, Chrysophyta, Cyanobacteria, Phaeophyceae, Rhodophyta |
| | Pteridophyta | Polypodiophyta |
| **Fungi** | Ascomycota | Saccharomycetes, Pezizomycetes, Eurotiomycetes, Lecanoromycetes, Leotiomycetes, Arthoniomycetes, Taphrinomycetes |
| | Basidiomycota | Basidiomycetes, Pucciniomycetes, Urediniomycetes, Entomophthoromycota, Tremellomycetes |
| **Bacteria** | Proteobacteria | Alphaproteobacteria, Betaproteobacteria, Gammaproteobacteria, Deltaproteobacteria, Epsilonproteobacteria |
| | Firmicutes | Bacilli, Clostridia, Tissierellia, Erysipelotrichia |
| | Actinobacteria | Actinobacteria, Coriobacteriia, Thermoleophilia, Acidimicrobiia, Rubrobacteria |
| | Spirochaetes | Spirochaetia, Leptospiria |
| | Cyanobacteria | Cyanobacteria |
| **Archaea** | Euryarchaeota | Methanobacteria, Methanococci, Methanopyri, Methanomicrobia, Halobacteria, Archaeoglobi, Thermoplasmata |
| | Korarchaeota | Korarchaeia |
| | Asgardarchaeota | Lokiarchaeia, Thorarchaeia, Heimdallarchaeia |
| | Crenarchaeota | Nitrososphaeria, Thermoprotei |

## C.2 MULTI-SPECIES GENOME FOR PRE-TRAINING

In this subsection, we introduce **GM-DATA**(train) used for pre-training. Table 10 groups representative species by their corresponding class and reports their genome sizes (Mb). By combining this with the hierarchical taxonomy presented in Table 9, each species can be mapped to its full taxonomic lineage. Covering a broad range of species and genome sizes, this dataset enables the model to capture cross-species evolutionary signals during pre-training, thereby improving cross-species generalization and learning biologically meaningful sequence representations.

Table 10: Representative species grouped by class with genome sizes (Mb).

| Class | Species | Genome Size (Mb) |
|---|---|---|
| Pucciniomycetes | *Puccinia striiformis* | 89.49 |
| | *Cronartium quercuum* | 76.57 |
| | *Puccinia graminis* | 88.72 |
| | *Uromyces viciae-fabae* | 215.71 |
| Basidiomycetes | *Fomitopsis palustris* | 35.25 |
| | *Lentinula edodes* | 45.59 |
| | *Armillaria mellea* | 70.85 |
| | *Pleurotus ostreatus* | 34.97 |
| | *Psilocybe cubensis* | 46.58 |
| Urediniomycetes | *Ustilago hordei* | 26.64 |
| | *Ustilago tritici* | 20.44 |
| | *Melanopsichium pennsylvanicum* | 21.12 |
| | *Sporisorium scitamineum* | 20.07 |
| Entomophthoromycota | *Beauveria bassiana* | 33.70 |
| | *Entomophthora muscae* | 260.30 |
| Tremellomycetes | *Tremella fuciformis* | 28.15 |
| | *Tremella mesenterica* | 28.64 |
| Eurotiomycetes | *Talaromyces marneffei* | 28.20 |

| Class | Species | Genome Size (Mb) |
|---|---|---|
| Eurotiomycetes | *Penicillium chrysogenum* | 32.41 |
| | *Aspergillus flavus* | 37.75 |
| | *Aspergillus niger* | 33.98 |
| Pezizomycetes | *Neurospora crassa* | 41.10 |
| | *Tuber melanosporum* | 124.95 |
| Lecanoromycetes | *Xanthoria parietina* | 30.44 |
| | *Evernia prunastri* | 40.35 |
| | *Cladonia grayi* | 26.09 |
| Leotiomycetes | *Monilinia fructicola* | 44.05 |
| | *Erysiphe necator* | 80.92 |
| | *Botrytis cinerea* | 42.63 |
| | *Sclerotinia sclerotiorum* | 38.46 |
| | *Podosphaera xanthii* | 152.75 |
| Saccharomycetes | *Candida albicans* | 14.28 |
| | *Yarrowia lipolytica* | 20.50 |
| | *Komagataella phaffii* | 9.22 |
| | *Prochlorococcus marinus* | 12.16 |
| Taphrinomycetes | *Taphrina deformans* | 13.34 |
| Arthoniomycetes | *Arthonia radiata* | 33.50 |
| Gastropoda | *Elysia chlorotica* | 260.17 |
| | *Littorina littorea* | 260.20 |
| | *Cornu aspersum* | 260.46 |
| | *Aplysia californica* | 260.45 |
| | *Conus consors* | 259.52 |
| Bivalvia | *Mytilus galloprovincialis* | 260.20 |
| | *Ruditapes philippinarum* | 259.79 |
| | *Sinanodonta woodiana* | 260.20 |
| | *Corbicula fluminea* | 0.66 |
| | *Magallana gigas* | 260.20 |
| | *Pinctada fucata* | 260.20 |
| Insecta | *Ctenocephalides felis* | 775.45 |
| | *Danaus plexippus* | 245.17 |
| | *Apis mellifera* | 225.25 |
| | *Aedes aegypti* | 1278.73 |
| | *Drosophila melanogaster* | 143.73 |
| Crustacea | *Pandalus borealis* | 260.13 |
| | *Procambarus clarkii* | 260.20 |
| | *Daphnia pulex* | 133.20 |
| | *Portunus trituberculatus* | 260.20 |
| | *Paralithodes camtschaticus* | 259.89 |
| Arachnida | *Pardosa pseudoannulata* | 260.46 |
| | *Parasteatoda tepidariorum* | 260.20 |
| | *Centruroides sculpturatus* | 260.43 |
| Aves | *Anser indicus* | 260.46 |
| | *Otis tarda* | 260.46 |
| | *Aix galericulata* | 260.46 |
| | *Aquila chrysaetos* | 260.20 |
| | *Columba livia* | 260.20 |
| | *Struthio camelus* | 260.20 |
| | *Cygnus cygnus* | 260.20 |
| | *Aquila rapax* | 260.20 |
| Reptilia | *Tiliqua scincoides* | 260.20 |
| | *Alligator sinensis* | 260.46 |
| | *Pogona vitticeps* | 260.46 |
| | *Crocodylus porosus* | 260.20 |
| | *Naja naja* | 260.20 |

| Class | Species | Genome Size (Mb) |
|---|---|---|
| Reptilia | *Testudo graeca* | 260.20 |
| | *Trachemys scripta elegans* | 260.20 |
| Amphibia | *Rana temporaria* | 260.46 |
| | *Aquarana catesbeiana* | 260.20 |
| | *Hyla sarda* | 260.20 |
| | *Bufo bufo* | 260.20 |
| Actinopterygii | *Gadus morhua* | 260.46 |
| | *Ictalurus punctatus* | 260.20 |
| | *Salmo salar* | 260.46 |
| | *Thunnus orientalis* | 260.20 |
| | *Oncorhynchus mykiss* | 260.20 |
| | *Astatotilapia calliptera* | 260.20 |
| | *Paralichthys olivaceus* | 260.46 |
| | *Cyprinus carpio* | 260.20 |
| | *Takifugu rubripes* | 260.20 |
| Mammalia | *Dog* | 260.20 |
| | *Bos taurus* | 260.20 |
| | *Ovis aries* | 260.20 |
| | *Balaenoptera musculus* | 260.20 |
| | *Equus caballus* | 260.20 |
| | *cat* | 260.20 |
| | *Rattus norvegicus* | 260.46 |
| | *Human* | 31372.10 |
| | *Macaca mulatta* | 260.20 |
| Bryopsida | *Mnium hornum* | 256.85 |
| | *Polytrichum commune* | 260.20 |
| Marchantiopsida | *Marchantia polymorpha* | 241.48 |
| | *Lunularia cruciata* | 260.20 |
| | *Riccia fluitans* | 260.46 |
| Polypodiophyta | *Adiantum capillus-veneris* | 260.46 |
| Magnoliopsida | *Malus domestica* | 260.46 |
| | *Brassica rapa* | 260.20 |
| | *Magnolia sinica* | 260.46 |
| | *Prunus mume* | 234.03 |
| | *Solanum lycopersicum* | 260.46 |
| | *Pisum sativum* | 260.46 |
| | *Glycine max* | 260.20 |
| | *Morus notabilis* | 259.05 |
| | *Strawberry* | 260.46 |
| Liliopsida | *Cucumis sativus* | 226.64 |
| | *Allium sativum* | 260.72 |
| | *Saccharum officinarum* | 260.20 |
| | *Secale cereale* | 260.20 |
| | *Zea mays* | 260.20 |
| | *Citrullus lanatus* | 260.20 |
| | *Lilium candidum* | 230.20 |
| | *Phalaenopsis equestris* | 260.17 |
| | *Cocos nucifera* | 260.20 |
| Phaeophyceae | *Ectocarpus siliculosus* | 195.81 |
| | *Saccharina japonica* | 260.20 |
| | *Macrocystis pyrifera* | 260.44 |
| Cyanobacteria | *Prochlorococcus marinus* | 1.70 |
| | *Nostoc punctiforme* | 9.06 |
| | *Nostoc sp. 7120* | 7.21 |

| Class | Species | Genome Size (Mb) |
|---|---|---|
| Cyanobacteria | *Synechocystis sp. 6803* | 3.95 |
| Chrysophyta | *Ochromonas danica* | 44.19 |
| | *Thalassiosira pseudonana* | 32.44 |
| | *Phaeodactylum tricornutum* | 27.45 |
| | *Fistulifera solaris* | 51.73 |
| Chlorophyta | *Ostreococcus tauri* | 13.03 |
| | *Chlamydomonas reinhardtii* | 111.10 |
| | *Volvox carteri* | 137.68 |
| | *Chlorella vulgaris* | 40.44 |
| Rhodophyta | *Cyanidioschyzon merolae* | 16.55 |
| | *Porphyra umbilicalis* | 87.89 |
| | *Chondrus crispus* | 104.98 |
| Betaproteobacteria | *Neisseria gonorrhoeae* | 2.17 |
| | *Vibrio cholerae* | 4.14 |
| | *Achromobacter xylosoxidans* | 6.90 |
| | *Burkholderia cepacia* | 8.37 |
| | *Zoogloea ramigera* | 4.61 |
| | *Bordetella pertussis* | 4.09 |
| | *Agrobacterium tumefaciens* | 6.00 |
| Epsilonproteobacteria | *Campylobacter jejuni* | 1.64 |
| | *Campylobacter coli* | 1.72 |
| | *CampylobactHelicobacter hepaticuser coli* | 1.80 |
| | *Campylobacter lari* | 1.49 |
| | *Helicobacter pylori* | 1.70 |
| Alphaproteobacteria | *Rhizobium leguminosarum* | 7.60 |
| | *Rhodospirillum rubrum* | 4.41 |
| | *Rickettsia rickettsii* | 1.26 |
| | *Bordetella pertussis* | 4.09 |
| | *Agrobacterium tumefaciens* | 6.00 |
| Gammaproteobacteria | *Salmonella enterica* | 4.95 |
| | *Vibrio cholerae* | 4.14 |
| | *Pseudomonas putida* | 6.16 |
| | *Haemophilus influenzae* | 1.89 |
| | *Pseudomonas aeruginosa* | 6.26 |
| | *Klebsiella pneumoniae* | 5.68 |
| | *Bordetella pertussis* | 4.09 |
| Deltaproteobacteria | *Desulfobacter postgatei* | 3.97 |
| | *Syntrophomonas wolfei* | 2.94 |
| | *Myxococcus xanthus* | 9.14 |
| | *Methanococcus maripaludis* | 1.71 |
| | *Geobacter sulfurreducens* | 3.81 |
| | *Sulfurimonas denitrificans* | 2.20 |
| Acidimicrobiia | *Acidimicrobium ferrooxidans* | 2.16 |
| | *Ilumatobacter fluminis* | 4.78 |
| Actinobacteria | *Corynebacterium glutamicum* | 3.31 |
| | *Frankia alni* | 7.50 |
| | *Corynebacterium diphtheriae* | 2.46 |
| | *Bifidobacterium longum* | 2.39 |
| | *Streptomyces coelicolor* | 8.67 |
| | *Mycobacterium leprae* | 3.19 |
| | *Mycobacterium tuberculosis* | 4.41 |
| Rubrobacteria | *Rubrobacter taiwanensis* | 3.04 |
| | *Rubrobacter radiotolerans* | 3.40 |
| | *Rubrobacter xylanophilus* | 3.23 |
| Thermoleophilia | *Conexibacter woesei* | 6.36 |

| Class | Species | Genome Size (Mb) |
|---|---|---|
| Thermoleophilia | *Solirubrobacter soli* | 9.31 |
| | *Thermoleophilum album* | 2.21 |
| | *Patulibacter minatonensis* | 5.52 |
| Coriobacteriia | *Eggerthella lenta* | 3.47 |
| | *Collinsella aerofaciens* | 2.46 |
| | *Gordonibacter pamelaeae* | 3.40 |
| | *Slackia heliotrinireducens* | 3.17 |
| Leptospiria | *Leptospira interrogans* | 5.52 |
| | *Leptospira noguchii* | 4.71 |
| | *Leptospira santarosai* | 3.98 |
| | *Leptospira biflexa* | 3.95 |
| | *Leptospira borgpetersenii* | 3.99 |
| | *Leptospira kirschneri* | 4.41 |
| Spirochaetia | *Treponema denticola* | 2.84 |
| | *Borreliella burgdorferi* | 1.32 |
| | *Brachyspira hyodysenteriae* | 3.09 |
| | *Spirochaeta thermophila* | 2.56 |
| | *Treponema pallidum* | 1.14 |
| Erysipelotrichia | *Erysipelothrix rhusiopathiae* | 1.79 |
| | *Turicibacter sanguinis* | 3.00 |
| | *Faecalicoccus pleomorphus* | 2.06 |
| | *Holdemania filiformis* | 3.74 |
| Tissierellia | *Helcococcus kunzii* | 2.10 |
| | *Peptoniphilus harei* | 1.93 |
| | *Finegoldia magna* | 1.84 |
| | *Parvimonas micra* | 1.68 |
| | *Anaerococcus vaginalis* | 1.89 |
| Clostridia | *Clostridium ljungdahlii* | 4.63 |
| | *Clostridium perfringens* | 3.36 |
| | *Selenomonas ruminantium* | 3.39 |
| | *Dialister invisus* | 1.90 |
| | *Megasphaera elsdenii* | 2.48 |
| | *Clostridium acetobutylicum* | 4.15 |
| | *Clostridium sporogenes* | 4.14 |
| | *Acidaminococcus fermentans* | 2.33 |
| | *Veillonella parvula* | 2.13 |
| | *Clostridium tetani* | 2.87 |
| | *Clostridioides difficile* | 4.10 |
| | *Clostridium botulinum* | 3.90 |
| | *Phascolarctobacterium succinatutens* | 2.35 |
| | *Clostridium beijerinckii* | 5.95 |
| Bacilli | *Geobacillus stearothermophilus* | 2.74 |
| | *Bacillus anthracis* | 5.50 |
| | *Paenibacillus polymyxa* | 5.97 |
| | *Bacillus thuringiensis* | 6.26 |
| | *Enterococcus faecalis* | 2.87 |
| | *Lactococcus lactis* | 2.64 |
| | *Lactobacillus acidophilus* | 1.98 |
| | *Staphylococcus aureus* | 2.82 |
| | *Listeria monocytogenes* | 2.94 |
| Halobacteria | *Halobacterium salinarum* | 2.43 |
| | *Haloferax volcanii* | 4.01 |
| | *Natronomonas pharaonis* | 2.75 |
| | *Halogeometricum borinquense* | 3.94 |
| | *Haloarcula marismortui* | 4.27 |
| | *Halorubrum lacusprofundi* | 3.69 |
| Methanomicrobia | *Methanosarcina mazei* | 4.14 |

| Class | Species | Genome Size (Mb) |
|---|---|---|
| Methanomicrobia | *Methanoculleus marisnigri* | 2.48 |
| | *Methanospirillum hungatei* | 3.54 |
| | *Methanosarcina barkeri* | 4.57 |
| | *Methanocorpusculum labreanum* | 1.80 |
| Methanopyri | *Methanopyrus kandleri* | 1.69 |
| Thermoplasmata | *Thermoplasma acidophilum* | 1.56 |
| | *Thermoplasma volcanium* | 1.58 |
| | *Ferroplasma acidarmanus* | 1.94 |
| | *Cuniculiplasma divulgatum* | 1.94 |
| Archaeoglobi | *Archaeoglobus veneficus* | 1.90 |
| | *Archaeoglobus profundus* | 1.56 |
| | *Archaeoglobus fulgidus* | 2.18 |
| Methanobacteria | *Methanothermobacter marburgensis* | 1.64 |
| | *Methanothermobacter thermautotrophicus* | 1.75 |
| | *Methanobacterium formicicum* | 2.49 |
| Methanococci | *Methanococcus voltae* | 1.78 |
| | *Methanocaldococcus jannaschii* | 1.74 |
| | *Methanococcus maripaludis* | 1.71 |
| | *Methanocaldococcus fervens* | 1.51 |
| Korarchaeia | *Candidatus Lokiarchaeum* | 1.59 |
| Nitrososphaeria | *Candidatus Nitrosopelagicus brevis* | 1.23 |
| | *Nitrosopumilus maritimus SCM1* | 1.65 |
| | *Candidatus Nitrososphaera evergl* | 2.95 |
| | *Nitrososphaera viennensis* | 2.53 |
| Thermoprotei | *Saccharolobus solfataricus* | 2.66 |
| | *Thermoproteus tenax* | 1.84 |
| | *Metallosphaera sedula* | 2.19 |
| | *Acidianus hospitalis* | 2.14 |
| | *Pyrobaculum aerophilum* | 2.22 |
| | *Saccharolobus islandicus* | 2.59 |
| Lokiarchaeia | *Candidatus Lokiarchaeum* | 1.59 |
| | *Promethearchaeum syntrophicum* | 4.32 |
| Heimdallarchaeia | *C. Heimdallarchaeota AB125* | 2.28 |
| | *C. Heimdallarchaeota LC3* | 5.68 |
| | *C. Heimdallarchaeota LC2* | 2.86 |
| Thorarchaeia | *C. Thorarchaeota SMTZ1-83* | 3.26 |

## C.3 MULTI-SPECIES GENOME FOR TESTING

In this subsection, we introduce **GM-DATA**(eval), an independent benchmark specifically designed to evaluate cross-species generalization. Table 11 lists the representative species included in **GM-DATA**(eval)along with their genome sizes (Mb). Constructed independently of the pre-training corpus, **GM-DATA**(eval)ensures a fair evaluation of the model's ability to generalize across unseen taxa. To guarantee balanced representation, all genomes were first segmented into 6,000 bp fragments with 100 bp overlaps. These sequences were then annotated at three hierarchical levels (Kingdom, Phylum, and Class), and 200 sequences were randomly sampled per species for each label set. This procedure yielded three stratified test sets containing 800, 2,000, and 3,000 samples, respectively. By covering 15 diverse species spanning multiple taxonomic classes, the benchmark captures both genomic complexity and taxonomic diversity, providing a rigorous environment for evaluating representation learning models in genomics.

## D LIMITATION

Although our dataset is more diverse and balanced than those used in DNABERT-2 and Nucleotide Transformer, it still covers only ∼300 species. This remains limited compared to the full biological

Table 11: Representative species and genome sizes used in the held-out test set.

| Kingdom | Phylum | Class | Species | Genome Size (Mb) |
|---|---|---|---|---|
| **Animalia** | Chordata | Actinopterygii | *Danio rerio* | 1448.79 |
| | | Mammalia | *Mus musculus* | 2728.22 |
| | | Amphibia | *Xenopus laevis* | 2742.47 |
| | | Aves | *Gallus gallus* | 1053.33 |
| | | Reptilia | *Chelonia mydas* | 2134.38 |
| **Bacteria** | Actinobacteria | Actinobacteria | *A. C. glutamicum* | 3.28 |
| | Firmicutes | Bacilli | *Bacillus subtilis* | 4.22 |
| | Proteobacteria | $\gamma$-Proteo | *Escherichia coli* | 4.64 |
| **Fungi** | Ascomycota | Saccharomycetes | *S. pastorianus* | 23.03 |
| | Basidiomycota | Basidiomycetes | *G. lucidum* | 47.52 |
| **Plantae** | Pteridophyta | Polypodiophyta | *C. richardii* | 7462.46 |
| | Gymnosperms | Ginkgoopsida | *Ginkgo biloba* | 2638.01 |
| | Bryophyta | Bryopsida | *P. patens* | 472.20 |
| | Angiosperms | Magnoliopsida | *A. thaliana* | 119.67 |
| | | Liliopsida | *Oryza sativa* | 386.34 |

diversity of nature, particularly for rare species and lineages that are yet to be sequenced. Consequently, the generalization ability of our model to such underrepresented groups may be constrained.

Moreover, while the proposed taxon-specific MoE demonstrates strong performance across the Domain → Kingdom → Phylum → Class hierarchy, extending this framework to finer-grained levels (e.g., Order, Family, Genus, and Species) would substantially increase model parameters and training complexity. This scalability challenge highlights the need for more efficient architectures and optimization strategies in future work.

## E  FUTURE WORKS

Future work will focus on the following directions:

- **Expanding data scale and species coverage:** Construct larger cross-species genomic corpora, with a particular emphasis on underrepresented groups such as Archaea, protists, and plants, to improve both generalization and fairness.
- **Extending to finer-grained taxonomic levels:** Adapt the MoE framework to deeper hierarchies (Order, Family, Genus, Species) to investigate the model's ability to capture subtle biological differences.
- **Improving computational efficiency:** Explore parameter sharing, sparse activation, and low-rank approximations to reduce the computational cost of MoE and enable more scalable training and deployment.

## F  USAGE OF LLM

In this work, large language models (LLMs) were used solely for writing assistance. Specifically, they were employed for polishing the language of the manuscript (e.g., grammar correction, clarity improvement, and bilingual translation when necessary). Importantly, LLMs were not involved in model design, dataset construction, training, evaluation, or analysis of results. All scientific contributions and experimental findings are entirely the work of the authors.

