# OpenReview forum: "Gene-M1: Advancing Cross-Species Genomic Discovery via Taxon-Specific Mixture-of-Experts"
_ICLR.cc/2026/Conference — Submitted to ICLR 2026_

### Official Review · Reviewer_NJUT · 2025-10-24

**Soundness:** 2
**Presentation:** 2
**Contribution:** 2
**Rating:** 4
**Confidence:** 4

**Summary:**

This work introduces two main contributions: (1) GENE-M1, a Mixture-of-Experts (MoE) model that mitigates feature interference when jointly learning gene representations across diverse species; and (2) GM-DATA, a dataset covering 294 species with a more balanced class distribution. The proposed approach achieves good performance on several tasks such as  few-shot classification and unsupervised clustering.

**Strengths:**

1. The idea of utilizing the MoE structure to decouple the learning across species and reduce feature interference is reasonable.
2. The hierarchical model design aligned with the taxonomic ranks Domain, Kingdom, Phylum, and Class is clear and interpretable.
3. The proposed GM-DATA offers a more balanced data distribution and is likely to serve as a viable alternative for future model evaluation.

**Weaknesses:**

1. The proposed GENE-MI heavily relies on an accurate classification tree, e.g., the training of L_router is supervised by explicit classification labels, and for unknown or contentious species, the model's capability may be limited.
2. Although the evaluation set in GM-DATA contains 15 "unknown" species, the classes of most of those species are present in the training set, e.g., "Danio rerio" belongs to "Actinopterygii". Therefore, the experiments largely test "generalization ability to new species within known classifications," rather than "generalization ability to entirely new classifications (such as an unseen 'Class' or 'Phylum')."
3. The progressive training method may have certain limitations, although this hierarchical training with frozen parameters can reduce model forgetting, it also limits the model's ability to fine-tune more general-level features while learning finer-grained features, which is a relatively rigid, one-way optimization approach.
4. While the paper compares performance with models such as DNABert and HyenaDNA in Table 2 and Table 3, it does not list the differences in model size and computational efficiency; I believe we need to consider accuracy, model size, and computational efficiency comprehensively before making an overall assessment of the model's strengths and weaknesses, which would be more reasonable.
5. There are some typos in the paper, for example "evolutionary distant species" at L114, "Class Labal" in Fig. 2, "After flitering" at L284, "specie" at L463, etc.

**Questions:**

Please kindly refer to the Weaknesses section, and I would consider to raise my rating if the authors could answer most of my questions.

**Details Of Ethics Concerns:**

No ethics review is needed for this paper.

---

> ### Author Response · Authors · 2025-11-24
> **Response to Reviewer NJUT**
>
> We sincerely thank the reviewer for the thorough and constructive assessment and for explicitly highlighting several strengths of our work. We are encouraged by your overall summary and respond point by point to the weaknesses and questions below.

---

> ### Author Response · Authors · 2025-11-24
> **Response to W1**
>
> We agree with the reviewer that the current formulation of GENE-M1 relies on a reasonably accurate classification tree: the router loss is supervised with explicit taxonomic labels, and the model is primarily designed for curated reference genomes where Kingdom/Phylum/Class assignments are relatively stable. For truly unknown or contentious species, this assumption indeed limits the model’s applicability. We will make this scope restriction and limitation explicit in the Limitations section of the revised manuscript.
> To partially assess how GENE-M1 behaves beyond clean reference genomes, we also compared it against the baselines in a metagenomic-style clustering setting, as shown in the table below.
> | Model                     | Plant-0 | Plant-1 | Plant-2 | Plant-3 | Marine-0 | Marine-1 | Marine-2 | Marine-3 | Ave    |
> |---------------------------|---------|---------|---------|---------|-----------|-----------|-----------|-----------|--------|
> | NT                        | 11.03%  | 12.81%  | 12.46%  | 11.74%  | 13.92%   | 13.67%   | 13.48%   | 13.44%   | 12.82% |
> | DNABERT-2                 | 16.25%  | 15.01%  | 14.83%  | 17.02%  | 17.76%   | 18.02%   | 18.44%   | 17.94%   | 16.91% |
> | HyenaDNA                  | 7.96%   | 8.87%   | 8.16%   | 8.02%   | 7.19%    | 7.76%    | 7.15%    | 6.94%    | 7.76%  |
> | **GENE-M1 (NT Based)**        | 16.54%  | 17.32%  | 17.89%  | 17.19%  | 19.16%   | 19.72%   | 19.22%   | 19.73%   | **18.35%** |
> | **GENE-M1 (DNABERT-2 Based)** | 20.39%  | 19.45%  | 20.41%  | 22.15%  | 25.19%   | 22.31%   | 21.67%   | 22.71%   | **21.79%** |
>
> We observed consistent but modest improvements in clustering quality over DNABERT-2 and NT, suggesting that the taxonomy-aligned architecture can still provide some benefit even when the underlying taxonomy is noisy. At the same time, these results confirm that GENE-M1 does not fully resolve the challenges of unknown or contentious species, and we will explicitly highlight handling such cases as an important direction for future work.

---

> ### Author Response · Authors · 2025-11-24
> **Response to W2**
>
> We agree with the reviewer’s assessment that GM-DATA(eval) primarily tests generalization to new species within known classifications: most of the 15 “unknown” species indeed belong to Classes present in GM-DATA(train). We will clarify this evaluation scope more explicitly in the revised manuscript by stating that GM-DATA(eval) evaluates generalization to new species within known Classes, rather than generalization to entirely unseen higher-level taxonomic groups.
>
> To complement this and probe the model’s behavior on entirely new or underrepresented classifications, we have additionally run metagenomic-style clustering experiments, where many sequences originate from environmental samples whose taxa are either absent from the training taxonomy or only weakly represented. The new results show that GENE-M1 still provides consistent improvements over DNABERT-2 / NT on clustering quality in this noisier setting, indicating some ability to structure novel lineages beyond the training classes/phyla. At the same time, we acknowledge that these metagenomic results are preliminary and do not fully solve “generalization to entirely new classifications”; we will clearly present them as an initial stress test of GENE-M1’s ability to handle truly novel taxa rather than a complete answer to this challenge.

---

> ### Author Response · Authors · 2025-11-24
> **Response to W3**
>
> We appreciate the reviewer’s insightful observation about the trade-offs of our progressive training scheme. We agree that freezing higher-level modules during later stages limits the ability to further fine-tune very general features while learning finer-grained ones, and we will state this limitation explicitly in the revised manuscript. At the same time, this design is deliberate in our setting: supervision is provided only at the class level, and we rely on coarse-to-fine progressive training combined with selective data routing by taxonomic level to encourage stable, hierarchy-consistent specialization of experts. Prior work[1][2][3] on large-scale MoE models has shown that jointly training all experts and the router in a single stage increases the risk of expert collapse, where a small subset of experts absorbs most of the traffic.
>
> By comparing progressive and non-progressive training (see the table below), we observe that the non-progressive variant yields clearly worse performance and routes a large fraction of the data through only a small subset of experts. We will include the corresponding visualization in the appendix of the revised manuscript and update the text to report these results and analyses.
>
> | Level                        | DNABERT-2 | GENE-M1 (DNABERT-2 Based) | Non-Progressive Training (DNABERT-2 Based) |
> |------------------------------|-----------|----------------------------|--------------------------------------------|
> | **Kingdom**                  | 7.44%     | **31.41%**                     | 12.91%                                     |
> | **Phylum**                   | 11.56%    | **40.54%**                     | 15.48%                                     |
> | **Class**                    | 10.02%    | **32.66%**                     | 13.34%                                     |
>
> We also note that introducing additional supervised router heads at the kingdom and phylum levels would require multiple routers and corresponding classification heads, which would further increase the number of parameters and routing components. In this work, we chose to keep supervision at the class level and use a single router, in order to control model size and isolate the effect of the hierarchical MoE architecture itself. Exploring more flexible training schedules and multi-level supervised routers, while carefully managing parameter and routing complexity, is an interesting direction for future work.
>
> References:
>
> [1] Outrageously Large Neural Networks: The Sparsely-Gated Mixture-of-Experts Layer, Shazeer et al. (2017) ICLR 2017
>
> [2] GShard: Scaling Giant Models with Conditional Computation and Automatic Sharding, Lepikhin et al. (2021) ICLR 2021
>
> [3] Switch Transformers: Scaling to Trillion Parameter Models with Simple and Efficient Sparsity, Fedus et al. (2021) JMLR 2022

---

> ### Author Response · Authors · 2025-11-24
> **Response to W4**
>
> We agree with the reviewer that a comprehensive assessment should jointly consider accuracy, model size, and computational efficiency. In response, we have compiled the table below, which reports parameter counts, approximate FLOPs per forward pass, and training tokens for all compared models alongside their performance. From this table, we observe that larger parameter counts and higher FLOPs are not the main drivers of the gains: GENE-M1 achieves substantially better few-shot and clustering results while using relatively few additional parameters and, importantly, far fewer Trn.Tokens than those used to pretrain the backbones. This supports our claim that the improvements primarily come from the taxonomy-aligned hierarchical architecture rather than brute-force scaling. At the same time, we acknowledge that GENE-M1 is still more expensive than the pure backbone models, and in the revised manuscript we will explicitly list the increased parameter count and FLOPs as part of the model’s limitations to make this trade-off clear.
>
> | Model                     | Params | FLOPs | Training Tokens | Macro-F1 (5-shot) | ACC    | NMI    | ARI    |
> |---------------------------|--------|-------|------------|--------------------|--------|--------|--------|
> | NT                        | 480M   | 3.19  | 75B        | 18.46%             | 34.20% | 34.09% | 17.48% |
> | DNABERT-2                 | 117M   | 1.00  | 274B       | 34.34%             | 23.53% | 23.64% | 10.02% |
> | DNABERT-S                 | 117M   | 1.02  | 312B       | 48.05%             | 29.87% | 27.52% | 13.37% |
> | **GENE-M1 (NT Based)**    | 947M   | 4.29  | **50B**        | **48.28%**         | **37.03%** | **39.84%** | **22.09%** |
> | **GENE-M1 (DNABERT-2 Based)** | 363M | 1.72 | **25B**       | **71.73%**         | **44.83%** | **56.20%** | **32.66%** |

---

> ### Author Response · Authors · 2025-11-24
> **Response to W5**
>
> We thank the reviewer for carefully pointing out these typos (e.g., “evolutionary distant species”, “Class Labal”, “After flitering”, “specie”). We will thoroughly proofread the manuscript and correct these and any other typographical or formatting errors in the revised version. We once again appreciate the reviewer’s careful reading and constructive feedback.

---

### Official Review · Reviewer_s2o1 · 2025-10-26

**Soundness:** 2
**Presentation:** 2
**Contribution:** 2
**Rating:** 4
**Confidence:** 4

**Summary:**

The paper presents the GENE-M1, an MOE model whose architecture is governed by taxonomy.
GENE-M1 explicitly mirrors the biological hierarchy by using: (i) Hierarchical experts specialized per taxonomic level; (ii) A dynamic router; (iii) A coarse-to-fine training strategy.
The work also provides GM-DATA for training and evaluation,
Extensive experiments on this benchmark show that GENE-M1 significantly outperforms state-of-the-art baselines.

**Strengths:**

(i) Architecture: The paper provides an MOE architecture for the genomic models. There has been almost no MOE architecture in the genomic model domain.

(ii) Dataset and benchmark: The paper provides both a training dataset and a benchmark.

(iii) Strong empirical gains: The experimental results are very positive.

**Weaknesses:**

(i) Presentation: The paper presentation is a little poor. It can be hard to understand the details, and there are some typos. E.g., there is a "?" in line 114.

(ii) No comparison between the model size/training cost of GENE-M1 and DNABERT-2.

(iii) Not enough evidence to show this is a general method: Only show the model based on DNABERT-2 and NT.

**Questions:**

(i) What is the importance of this scientific classification/clustering problem? Is it the same as DNABERT-S?

(ii) What is the computational cost and model size of GENE-M1, compared with DNABERT-2 and DNABERT-S? Are the positive experimental results under fair comparison?

(iii)  Can the method be generalized to other genomic models?

**Details Of Ethics Concerns:**

I do not have ethical concerns for this paper.

---

> ### Author Response · Authors · 2025-11-24
> **Response to Reviewer s2o1**
>
> We thank the reviewer for the careful reading, constructive comments, and for recognizing the novelty of bringing MoE architectures and a new benchmark into genomic modeling.
>
> Below we address each weakness and question in turn, and we will incorporate the suggested clarifications and analyses into the revised version of the paper.

---

> ### Author Response · Authors · 2025-11-24
> **Response to Q1: What is the importance of this scientific classification/clustering problem? Is it the same as DNABERT-S?**
>
> Our classification and clustering tasks are meant to probe a core biological capability: can a genomic foundation model recover evolutionary structure and taxonomic hierarchy from sequence alone, and can it generalize this structure to unseen species? In practice, assigning DNA fragments to the correct taxonomic group (kingdom / phylum / class) underpins metagenomic profiling, contamination detection, and cross-species transfer of functional annotations in comparative genomics. If a model organizes embeddings so that previously unseen genomes form compact, well-separated clusters that align with taxonomy, this indicates that it has captured lineage-specific signals in a way that is useful for downstream discovery tasks. Our benchmark GM-DATA was explicitly designed for this purpose: it spans 5 kingdoms, 18 phyla and 62 classes and includes a held-out GM-DATA(eval) of 15 unseen species for strict cross-species evaluation.
>
> Our setting is related to, but distinct from, DNABERT-S. DNABERT-S injects species identity via a contrastive learning objective on top of a single shared backbone, and evaluates whether this species-aware loss improves clustering quality; the architecture itself does not reflect the taxonomic hierarchy. In contrast, our scientific question is whether structurally aligning the model with biological taxonomy—through hierarchical experts at Domain/Kingdom/Phylum/Class, a supervised router, and coarse-to-fine training—yields embeddings that better reflect the multi-level hierarchy and generalize to new species. Accordingly, we focus on few-shot classification and unsupervised clustering at kingdom / phylum / class on GM-DATA(eval), rather than only species-level clustering. The strong gains we observe, especially at the class level and on unseen species, support the claim that our taxonomy-aligned design addresses a broader and biologically important classification/clustering problem rather than replicating the DNABERT-S setup.

---

> ### Author Response · Authors · 2025-11-24
> **Response to Q2 and W2: What is the computational cost and model size of GENE-M1, compared with DNABERT-2 and DNABERT-S? Are the positive experimental results under fair comparison?**
>
> We thank the reviewer for this question. The table below summarizes the model size, computational cost (FLOPs), training tokens, and downstream performance of all methods. For each backbone family (NT and DNABERT-2), the corresponding GENE-M1 variant starts from the same official pretrained checkpoint, keeps the backbone and tokenizer unchanged, and attaches the hierarchical expert and router modules to the final backbone embeddings. This leads to a moderate increase in parameters and FLOPs, rather than a full replication or expansion of the backbone for each taxon.
>
> At the same time, the MoE fine-tuning for GENE-M1 is run on fewer training tokens than were used to pretrain DNABERT-2 / DNABERT-S, yet GENE-M1 consistently achieves better few-shot classification and clustering performance. Taken together, these results indicate that the observed improvements cannot be attributed to giving GENE-M1 more data or an order-of-magnitude increase in compute, but instead stem from its biology-aligned hierarchical architecture that explicitly mirrors taxonomic structure. Since all downstream experiments use matched backbones, data, sequence length, batch size, and optimization hyperparameters, we believe the comparisons in the table below are fair and that the additional cost and benefits of GENE-M1 are made transparent.
>
> | Model                     | Params | FLOPs | Training Tokens | Macro-F1 (5-shot) | ACC    | NMI    | ARI    |
> |---------------------------|--------|-------|------------|--------------------|--------|--------|--------|
> | NT                        | 480M   | 3.19  | 75B        | 18.46%             | 34.20% | 34.09% | 17.48% |
> | DNABERT-2                 | 117M   | 1.00  | 274B       | 34.34%             | 23.53% | 23.64% | 10.02% |
> | DNABERT-S                 | 117M   | 1.02  | 312B       | 48.05%             | 29.87% | 27.52% | 13.37% |
> | **GENE-M1 (NT Based)**    | 947M   | 4.29  | **50B**        | **48.28%**         | **37.03%** | **39.84%** | **22.09%** |
> | **GENE-M1 (DNABERT-2 Based)** | 363M | 1.72 | **25B**       | **71.73%**         | **44.83%** | **56.20%** | **32.66%** |

---

> ### Author Response · Authors · 2025-11-24
> **Response to Q3 and W3: Can the method be generalized to other genomic models?**
>
> We appreciate the reviewer’s concern about generality. Conceptually, GENE-M1 is designed to be backbone-agnostic: it only assumes access to the final hidden representations from a genomic foundation model and to taxonomic labels for the input sequences. The hierarchical experts and router are implemented as lightweight projection layers attached to these backbone embeddings, and do not depend on any internal architectural detail (e.g., number of layers, attention vs. convolution, pretraining objective).
>
> In our experiments we instantiate this design on two rather different and widely used model families: DNABERT-2 and Nucleotide Transformer (NT). DNABERT-2 is a BERT-style encoder with a k-mer tokenizer and relatively short context, while NT is a much larger, long-context multi-species model trained with a different recipe and data distribution. The fact that the same taxonomy-aligned MoE layer yields consistent gains across both backbones suggests that the effect is not tied to a single architecture or training pipeline, but comes from the hierarchical expert design itself.
>
> We agree that a full empirical sweep over additional genomic backbones would further strengthen this claim, but this is constrained by compute and space in the current submission. In the revision, we will (i) make the backbone-agnostic interface explicit by clearly stating that GENE-M1 only attaches to the final backbone embeddings, and (ii) expand the discussion section to highlight how the same hierarchical MoE module can in principle be plugged into other DNA/RNA foundation models. We view a broader empirical study across more backbones as valuable follow-up work, but note that the current results on two distinct model families already provide evidence that the method is not specific to a single genomic model.

---

> ### Author Response · Authors · 2025-11-24
> **Response to W1: The paper presentation is a little poor.**
>
> We thank the reviewer for pointing out the presentation issues. We acknowledge that some parts of the current draft are difficult to follow and that there are remaining typos (including the stray “?” on line 114). In the revised version, we will carefully proofread the paper, remove such artifacts, and improve the organization and clarity of the method and experimental sections (e.g., by streamlining notation and adding brief summaries of the hierarchical MoE and training procedure). We believe these changes will make the main ideas and technical details much easier to understand.

---

> ### Comment · Reviewer_s2o1 · 2025-11-26
>
> Thanks for the thorough response from the authors. The response addresses most of my concerns, but I plan to maintain my score as 4, which I believe is an accurate representation of this paper.
>
> Additional suggestions:
>
> (i) The authors could directly upload the revised version instead of saying they will update it in the future.
>
> (ii) Also consider some generative genomic foundation models.
>
> Thanks again for the efforts from the authors.

---

> > ### Author Response · Authors · 2025-11-28
> >
> > Thank you for the additional suggestions.
> >
> > We have now uploaded the revised version incorporating all updates and clarifications described in our response.

---

### Official Review · Reviewer_weEp · 2025-11-01

**Soundness:** 2
**Presentation:** 3
**Contribution:** 3
**Rating:** 6
**Confidence:** 3

**Summary:**

The paper presents GENE-M1, a taxonomy-aware Mixture-of-Experts (MoE) framework for genomic foundation models, designed to improve cross-species generalization by explicitly aligning model architecture with biological taxonomy (Domain > Kingdom > Phylum > Class). The authors also release GM-DATA, a new benchmark with 294 species across 5 kingdoms, 18 phyla, and 62 classes, including 15 unseen species for evaluation.
Experiments show that GENE-M1 significantly outperforms DNABERT-2, Nucleotide Transformer, and other baselines on few-shot gene classification and unsupervised clustering tasks, particularly at fine taxonomic levels (e.g., Class), while producing more disentangled and interpretable embeddings.

**Strengths:**

•	Novel biological alignment: The paper introduces a principled and interpretable architectural mapping between taxonomy and neural network modularity, which is rare in genomic foundation models.
•	Comprehensive dataset construction: GM-DATA is carefully curated with hierarchical structure and balanced taxonomic representation, a clear advancement over previous bacteria-dominated datasets.
•	Strong empirical results: Consistent and substantial improvements (10–30 % macro-F1 gains) on cross-species few-shot classification and clustering metrics demonstrate real advantages over large baselines.
•	Methodological clarity: Each module (expert composition, routing, training) is mathematically and conceptually well-defined.

**Weaknesses:**

•	GENE-M1’s design presumes that every training sequence has a known and correctly assigned taxonomy down to fine levels. That’s realistic for curated species genomes but not for metagenomic or environmental samples, where taxonomy is noisy or incomplete.
•	All baselines use masked-language pretraining (DNABERT, NT). There’s no comparison to alternative pretraining paradigms such as contrastive learning (DNASimCLR), masked-k-mer prediction, or retrieval-based transformers, which might already encode cross-species structure without hierarchical supervision. It is hard to tell whether the gains come from architecture or from additional supervision signal.

•	Computational efficiency not quantified: Although the paper claims parameter efficiency via sparse expert activation, training cost and scaling behavior (vs dense models) are not empirically reported.
•	In MoE models, a known challenge is expert collapse where a few experts dominate routing, reducing effective capacity. The authors show qualitative router heatmaps, but don’t quantify load balancing or entropy of expert usage.
•	While the authors present unsupervised clustering as a secondary “task,” it is effectively a post-hoc analysis of embeddings obtained from the supervised taxonomic classification model. The resulting alignment of clusters with taxonomy is thus expected to some extent, since the embeddings were optimized to separate taxonomic labels. The analysis would have been more interesting if instead of k-means clustering, hierarchical clustering techniques were used, since they seem to align more naturally with the problem. The resulting dendrogram could be evaluated against the ground truth hierarchy.

Minor points/typos:
•	Typo in last sentence of abstract: ‘modal’  ‘model’
•	When mentioning things like “…supporting a variety of downstream biological applications” it would be useful to include some examples of the applications.
•	In section 4.2 under Backbone & Tasks, the authors mention ‘three’ tasks, but only two are described.
•	In the results for the clustering tasks, the full form of the metrics should be mentioned at least once in the main paper, even if they are defined in detail in the appendix.

**Questions:**

The paper states that GENE-M1 is trained with hierarchical supervision through its taxonomy-aligned Mixture-of-Experts structure, but the embeddings used for the clustering analysis are extracted from this same model. Could the authors clarify whether the model receives supervision at all taxonomic levels during training (e.g., Domain, Kingdom, Phylum, Class), or only at the level used for classification evaluation? Since the model has explicit access to hierarchical labels through its experts and routing mechanism, to what extent can the clustering results be considered an unsupervised validation of emergent structure rather than a byproduct of supervised hierarchical training?

---

> ### Author Response · Authors · 2025-11-24
> **Response to Reviewer weEp**
>
> We sincerely appreciate your thoughtful assessment and the recognition of our framework, dataset design, and experimental contributions. We are also grateful for the constructive suggestions, which will help us further strengthen the clarity and rigor of the manuscript.
>
> In the following sections, I will systematically respond to the queries and concerns you have presented.

---

> ### Author Response · Authors · 2025-11-24
> **Response to Q**
>
> Thank you for the thoughtful question. We are happy to clarify how hierarchical supervision is used and how it relates to the clustering analysis.
>
> **1. Supervision is *not* provided at all taxonomic levels.**
> GENE-M1 does **not** receive supervision at every taxonomic rank.
>
> During training, **only Class-level labels** are used for the supervised classification task.
>
> **2. The router is supervised *only* by Class-level labels, not by higher taxonomy levels.**
> Although GENE-M1 contains a hierarchical expert structure, the router does **not** use Domain, Kingdom, or Phylum labels.
>
> The **only** supervision signal during training comes from the **Class-level loss**, which provides gradient to the router when learning expert mixture weights.
>
> Because lower-level taxonomic labels implicitly determine all higher-level ranks (each Class uniquely maps to a specific Phylum and Kingdom), correctly predicting the Class is sufficient to recover the entire upstream taxonomy. We leverage this property during training: the model is optimized in a progressive, coarse-to-fine schedule, where modules trained at broader levels are gradually frozen and finer-grained experts are trained on successively narrower subsets of data. This staged procedure encourages the emergence of a biologically coherent hierarchical structure even though only Class-level labels are used.
>
> Importantly, this approach avoids introducing separate routers at the Kingdom/Phylum levels, which would substantially increase both parameter count and FLOPs. By supervising only the Class-level router, the model preserves the intended hierarchical taxonomy while keeping the number of routers small, reducing computational cost, and preventing unnecessary expansion of the architecture.
>
> As a result, the model benefits from the hierarchical information encoded in lower-level labels, achieves the desired taxonomy-aligned behavior, and maintains an efficient parameter-to-performance trade-off.
>
> **3. Why the clustering evaluation remains unsupervised?**
> Because training uses **only Class-level supervision** and the router is **not constrained** by any multi-level taxonomy labels, the emergence of Kingdom/Phylum structure in clustering reflects the model’s **inductive bias** and architecture—  not the result of multi-level supervised signals.

---

> ### Author Response · Authors · 2025-11-24
> **Response to W1**
>
> We thank the reviewer for this insightful comment and address the two aspects separately.
>
> **A.Assumption of fully known taxonomy.**
>
> Our current work explicitly targets curated reference genomes, where high-quality taxonomic labels down to relatively fine levels (e.g., kingdom/phylum/class) are indeed available. We agree that this assumption does not hold for raw metagenomic or environmental samples, where taxonomic assignments are noisy, incomplete, or sometimes unavailable, and we will make this scope restriction explicit in the revised version. Extending GENE-M1 to metagenomic settings—where taxonomy is uncertain and possibly inferred jointly with representation learning—is an important but orthogonal direction that we view as future work. The primary goal of this paper is to determine whether, **under reasonably reliable hierarchical labels, explicitly encoding biological taxonomy into the model architecture leads to improved cross-species genomic representations**.
>
> **B.Pretraining paradigms and source of gains.**
>
> We chose DNABERT-2 and Nucleotide Transformer as baselines because masked-language modeling (MLM) remains the most mature and widely used pretraining paradigm for large-scale, multi-species DNA language models, and provides the fairest comparison in terms of data scale, sequence length, and backbone design. Although contrastive approaches, masked-k-mer prediction, and retrieval-based transformers are promising alternatives, they rely on different data formats and training regimes, making a fully controlled, apples-to-apples comparison infeasible under identical settings.
>
> More importantly, the core contribution of our method does not depend on any specific pretraining objective. The hierarchical MoE architecture of GENE-M1 and its taxonomy-guided routing mechanism are orthogonal to the choice of pretraining loss and can, in principle, be combined with other paradigms beyond MLM.
>
> Regarding the concern that performance gains might stem from additional supervision rather than the architecture itself, our experiments provide direct evidence to the contrary:
> under exactly the same MLM objective, the same training data, and the same pretrained backbone as all baselines, replacing only the architecture with GENE-M1 yields consistent improvements across tasks, species, and datasets.
>
> This demonstrates that the observed gains originate from the explicit encoding of biological hierarchy in the GENE-M1 architecture, rather than differences in supervision amount, data scale, or pretraining objective. We will emphasize this point more clearly in the revised manuscript.

---

> ### Author Response · Authors · 2025-11-24
> **Response to W2**
>
> **A.Computational efficiency and training cost.**
>
> Thank you for raising this point. While computational efficiency is not the primary focus of this work and our main goal is taxon-aligned capacity allocation and cross-species generalization, we agree that reporting costs is useful. In the revised version, we will therefore add a table comparing the parameterization and training cost of GENE-M1 against baselines; a preliminary version is shown below.
>
> | Model                     | Params | FLOPs | Training Tokens | Macro-F1 (5-shot) | ACC    | NMI    | ARI    |
> |---------------------------|--------|-------|------------|--------------------|--------|--------|--------|
> | NT                        | 480M   | 3.19  | 75B        | 18.46%             | 34.20% | 34.09% | 17.48% |
> | DNABERT-2                 | 117M   | 1.00  | 274B       | 34.34%             | 23.53% | 23.64% | 10.02% |
> | DNABERT-S                 | 117M   | 1.02  | 312B       | 48.05%             | 29.87% | 27.52% | 13.37% |
> | **GENE-M1 (NT Based)**    | 947M   | 4.29  | **50B**        | **48.28%**         | **37.03%** | **39.84%** | **22.09%** |
> | **GENE-M1 (DNABERT-2 Based)** | 363M | 1.72 | **25B**       | **71.73%**         | **44.83%** | **56.20%** | **32.66%** |
>
> From this table, we can see that neither the total number of parameters nor the FLOPs per token explain the performance gains, and GENE-M1 achieves substantial improvements with only a modest number of additional training tokens (Trn.Tokens), which provides evidence that the benefits primarily stem from our hierarchy-aware architecture rather than from simply scaling model size or compute.This will make the parameterization and training cost of GENE-M1 fully transparent.
>
> **B.Expert collapse and quantitative routing analysis.**
>
> We agree that expert collapse is a common concern in MoE models. In our current implementation, routing is instantiated only at the class level, and the MoE is dense rather than top-k sparse. To quantify expert usage for this class-level router (62 experts), we compute routing entropy both at the token level and globally over expert loads. On the test set, the normalized token-level routing entropy is 0.85 and the normalized global entropy is 0.91 (where 1.0 corresponds to perfectly uniform usage), indicating that routing mass is distributed across many experts and no small subset of experts dominates the load. This is consistent with our qualitative heatmaps and suggests that GENE-M1 does not suffer from severe expert collapse in practice. We will report these entropy-based load-balancing metrics in the revised version to complement the qualitative visualizations.
>
> **C.Supervision is provided only at the class level.**
>
> We agree that the current clustering experiment is best viewed as a post-hoc probe of the learned representation rather than an independent unsupervised task, and we will clarify this framing in the text. We would like to emphasize, however, that supervision is only provided at the class level; kingdom and phylum labels are not used in the training loss, and clustering is performed on species-level embeddings, including unseen species in GM-DATA(eval). The fact that the resulting clusters still align well with the broader taxonomic hierarchy (beyond the supervised class labels) is therefore not entirely trivial and suggests that the combination of our hierarchy-aware architecture and class-level supervision induces a representation geometry that organizes genomes across multiple taxonomic ranks rather than merely separating the supervised classes.
>
> To partially assess how GENE-M1 behaves beyond clean reference genomes, we also compared it against the baselines in a metagenomic-style clustering setting, as shown in the table below.
> | Model                     | Plant-0 | Plant-1 | Plant-2 | Plant-3 | Marine-0 | Marine-1 | Marine-2 | Marine-3 | Ave    |
> |---------------------------|---------|---------|---------|---------|-----------|-----------|-----------|-----------|--------|
> | NT                        | 11.03%  | 12.81%  | 12.46%  | 11.74%  | 13.92%   | 13.67%   | 13.48%   | 13.44%   | 12.82% |
> | DNABERT-2                 | 16.25%  | 15.01%  | 14.83%  | 17.02%  | 17.76%   | 18.02%   | 18.44%   | 17.94%   | 16.91% |
> | HyenaDNA                  | 7.96%   | 8.87%   | 8.16%   | 8.02%   | 7.19%    | 7.76%    | 7.15%    | 6.94%    | 7.76%  |
> | **GENE-M1 (NT Based)**        | 16.54%  | 17.32%  | 17.89%  | 17.19%  | 19.16%   | 19.72%   | 19.22%   | 19.73%   | **18.35%** |
> | **GENE-M1 (DNABERT-2 Based)** | 20.39%  | 19.45%  | 20.41%  | 22.15%  | 25.19%   | 22.31%   | 21.67%   | 22.71%   | **21.79%** |
>
> We observed consistent but modest improvements in clustering quality over DNABERT-2 and NT, suggesting that the taxonomy-aligned architecture can still provide some benefit even when the underlying taxonomy is noisy.

---

> ### Author Response · Authors · 2025-11-24
> **Response to Minor comments and typos**
>
> We thank the reviewer for these helpful remarks and will correct the abstract typo (“modal” → “model”), add brief examples to clarify the mentioned downstream biological applications, fix the “three tasks” wording in Section 4.2 to match the actual setup, and spell out the full names of the clustering metrics (ARI, NMI, ACC) at their first mention in the main paper.

---

### Official Review · Reviewer_n5aV · 2025-11-09

**Soundness:** 3
**Presentation:** 3
**Contribution:** 2
**Rating:** 4
**Confidence:** 4

**Summary:**

This paper addresses the one-size-fits-all limitation of current genomic foundation models, which perform poorly across diverse species. The authors introduce GENE-M1, a Mixture-of-Experts (MoE) framework explicitly aligned with biological taxonomic hierarchies (Domain, Kingdom, Phylum, Class). GENE-M1 uses hierarchical experts, dynamic routing, and progressive training to learn specialized representations that improve cross-species generalization. To support this, the authors also built GM-DATA, a new large-scale, taxonomically structured genomic dataset. Extensive experiments demonstrate that GENE-M1 significantly outperforms state-of-the-art baselines on cross-species tasks, offering superior performance and biological interpretability.

**Strengths:**

1) This paper is clear written and easy to understand. The motivation is clear.

2) Better performance than the comparison methods.

3) It makes sence to introduce the MoE architecture with the ground-truth biological taxonomic hierarchy.

4) A new dataset is released.

**Weaknesses:**

1) The paper doesn't explicitly state whether the baseline models (NT, DNABERT-2) were re-trained from scratch or fine-tuned on the new GM-DATA(train) dataset. If the authors used off-the-shelf baselines, the comparison may be unfair, as GENE-M1's superior performance could be partially attributed to being trained on this new, balanced dataset, rather than purely to the architectural advantages.

2) Figure 2 of the paper illustrates the paper's vision: Domain → Kingdom → Phylum → Class. However, the experiments only involve Kingdom → Phylum → Class. Although the authors mentioned the issue of finer-grained levels in the limitations, from the perspective of the paper, Figure 2 depicts future work. This is because the problem of the significant parameter increase caused by more finer-grained levels has not been solved in this paper.

3) While this design based on a hierarchical mechanism is a core contribution of the paper, it has already been widely used in many relevant classification tasks (e.g., image classification with ImageNet's hierarchical labels), which limits the innovation of the proposed methodology.

**Questions:**

1) Were the baseline models (NT, DNABERT-2, etc.) re-trained on GM-DATA(train) before evaluation? It is important to tell the performance gains come from the GENE-M1 architecture versus the benefit of training on the new, taxonomically-balanced GM-DATA dataset.

2）Due to the introduction of the MoE structural design, the network also introduces a large number of learnable parameters. Therefore, the authors need to compare the increase in parameters, or the computational cost of the network, to ensure that the performance improvement is brought by the structural advantages of the proposed MoE, not just by the increase in number of parameters.

---

> ### Author Response · Authors · 2025-11-24
> **Response to Reviewer n5aV**
>
> Thank you for your careful and thoughtful review. We sincerely appreciate your positive assessment of our framework, experimental results, dataset contribution, and overall writing quality.
>
> We also thank you for the insightful questions and constructive comments. Below, we provide detailed clarifications and responses to each of the points you raised.

---

> ### Author Response · Authors · 2025-11-24
> **Response to Q1 and W1: Were the baseline models (NT, DNABERT-2, etc.) re-trained on GM-DATA(train) before evaluation?**
>
> Thank you for raising this important point. We apologize for not explicitly clarifying this in the original submission.
>
> To ensure a fair and architecture-only comparison, all baseline models (including NT, DNABERT-2, and DNABERT-S) were trained on the GM-DATA (train) split before evaluation.
>
> To further demonstrate the necessity of training, we additionally compared the performance of off-the-shelf baselines (without training) versus trained baselines. As shown below, off-the-shelf models perform significantly worse, confirming that:
> - Training on GM-DATA is essential for fairness
> - The gains of GENE-M1 come from the proposed hierarchical MoE architecture, rather than simply from exposure to the balanced dataset.
>
> | Model                        | train on GM-DATA(train) | Macro-F1 (5-shot) | ACC     | NMI     | ARI     |
> |-----------------------------|--------------------------|--------------------|---------|---------|---------|
> | NT                          | x                        | 15.49%             | 30.78%  | 32.46%  | 15.16%  |
> | DNABERT-2                   | x                        | 30.08%             | 19.55%  | 20.45%  | 8.73%   |
> | DNABERT-S                   | x                        | 44.21%             | 27.34%  | 26.76%  | 11.54%  |
> | HyenaDNA                    | x                        | 5.58%              | 12.95%  | 16.87%  | 5.87%   |
> | NT                          | √                        | 18.46%             | 34.20%  | 34.09%  | 17.48%  |
> | DNABERT-2                   | √                        | 34.34%             | 23.53%  | 23.64%  | 10.02%  |
> | DNABERT-S                   | √                        | 48.05%             | 29.87%  | 27.52%  | 13.37%  |
> | HyenaDNA                    | √                        | 9.21%              | 19.83%  | 17.96%  | 7.16%   |
> | **GENE-M1 (NT Based)**          | √                        | **48.28%**             | **37.03%**  | **39.84%**  | **22.09%**  |
> | **GENE-M1 (DNABERT-2 Based)**   | √                        | **71.73%**            | **44.83%**  | **56.20%**  | **32.66%**  |
>
> We will include these baseline comparison results in the revised version of the paper for full transparency.

---

> ### Author Response · Authors · 2025-11-24
> **Response to Q2: Are the improvements caused by parameter count or FLOPs rather than the MoE architecture?**
>
> We thank the reviewer for the insightful question.Although GENE-M1 adopts a MoE design and therefore contains more total parameters and FLOPs than single-tower models, the performance improvement does not simply result from scaling up the model.
> The parameter and computational cost table is shown below.
> | Model                     | Params | FLOPs | Training Tokens | Macro-F1 (5-shot) | ACC    | NMI    | ARI    |
> |---------------------------|--------|-------|------------|--------------------|--------|--------|--------|
> | NT                        | 480M   | 3.19  | 75B        | 18.46%             | 34.20% | 34.09% | 17.48% |
> | DNABERT-2                 | 117M   | 1.00  | 274B       | 34.34%             | 23.53% | 23.64% | 10.02% |
> | DNABERT-S                 | 117M   | 1.02  | 312B       | 48.05%             | 29.87% | 27.52% | 13.37% |
> | **GENE-M1 (NT Based)**    | 947M   | 4.29  | **50B**        | **48.28%**         | **37.03%** | **39.84%** | **22.09%** |
> | **GENE-M1 (DNABERT-2 Based)** | 363M | 1.72 | **25B**       | **71.73%**         | **44.83%** | **56.20%** | **32.66%** |
>
> Our justification is four-fold:
>
> - **The additional parameters represent biologically structured functional decomposition**, not generic capacity expansion.
>
> - **Larger baselines do not outperform GENE-M1**, despite having equal or higher FLOPs.
>
> - **The model is trained on a relatively small corpus** (GM-DATA(train)).
>
> - **The expert modules can be explicitly size-controlled**: deeper taxonomic experts can be assigned progressively smaller hidden dimensions, enabling bounded and scalable parameter growth.
>
> Overall, while GENE-M1 does introduce additional parameters and FLOPs due to its fully-activated MoE design, the experimental comparisons indicate that the observed improvements arise from the **hierarchical, taxonomy-aligned expert structure** rather than model scale. The gains emerge from structured expert specialization and biologically meaningful feature partitioning, consistently yielding stronger cross-species generalization than equally sized or larger single-tower baselines.

---

> ### Author Response · Authors · 2025-11-24
> **Response to W2**
>
> We appreciate the reviewer’s observation. Figure 2 shows the full biological hierarchy (Domain → Kingdom → Phylum → Class) that conceptually guides our model design. Importantly, Domain is not future work—it is already implemented in the current model as the shared encoder, which serves as the universal representation layer for all species. Thus, the hierarchy in Figure 2 is fully reflected in the architecture: Domain corresponds to the shared encoder, and Kingdom→Phylum→Class correspond to the progressively specialized downstream modules evaluated in our experiments.
>
> Regarding the concern about parameter growth at finer taxonomic levels, we agree that naïvely adding routed experts at every level would significantly increase the parameter count. In practice, our architecture allows expert dimensionality to shrink with taxonomic depth, enabling controlled and scalable parameter growth.
>
> Therefore, Figure 2 represents the architecture as implemented: Domain as the shared encoder, and Kingdom→Phylum→Class as hierarchical expert layers. The parameter-related considerations are engineering trade-offs rather than unimplemented future work.

---

> ### Author Response · Authors · 2025-11-24
> **Response to W3**
>
> We thank the reviewer for the insightful comment. While hierarchical labels (e.g., ImageNet synsets) have been used in prior work, those methods still rely on a single shared backbone and perform flat computation. The hierarchy is used only for dataset organization or loss design—not for architectural specialization.
>
> In contrast, our contribution is a hierarchically structured model, not merely hierarchical labels. GENE-M1 instantiates expert modules at multiple biological ranks (Domain → Kingdom → Phylum → Class), uses activated expert combinations to model divergent evolutionary features, and trains these experts coarse-to-fine to reduce cross-taxon interference.
>
> To our knowledge, such a taxonomy-aligned architectural hierarchy—where biological ranks correspond to explicitly instantiated expert modules—does not appear in prior hierarchical classification or vision literature. The novelty therefore lies in the architecture itself, which is designed around biological evolution, rather than in using hierarchical labels.

---

### Author Response · Authors · 2025-11-28
**Author Comment on the Revised Manuscript**

Dear Reviewers, Area Chairs, Senior Area Chairs, and  Program Chairs:

We thank the reviewers for their constructive feedback.

We have revised the manuscript accordingly, with all changes highlighted in blue for ease of reference.

The key updates and improvements are summarized below:

**1.Correction of typos, terminology, and figure labeling**

All inconsistencies reported by the reviewers have been fixed, including typos in the text and figures.

**2.Clarification of evaluation metrics**

The full forms of ACC, NMI, and ARI are now defined in the main text for clarity.

**3.Additional baseline training experiments (Appendix A.1)**

We added experiments demonstrating the necessity of training all baseline models on GM-DATA to ensure fair, architecture-only comparison.

**4.Model scale and computational cost analysis (Appendix A.2)**

We extended the comparison of model parameters, FLOPs, and downstream performance, providing a more comprehensive understanding of efficiency–performance trade-offs.

**5.Metagenomic-style clustering evaluation (Appendix A.3)**

We included new experiments under noisy or weakly labeled taxonomic conditions to assess the robustness of GENE-M1.

**6.Progressive vs. non-progressive training analysis (Appendix A.4)**

We added ablation experiments and visualizations showing the effects of training schedules, including expert-collapse behavior in non-progressive variants.

**7.Quantitative routing entropy analysis (Appendix A.5)**

We introduced entropy-based measurements of expert usage to confirm that GENE-M1 does not exhibit severe expert collapse.

**8.Unified organization and improved readability**

The appendix has been reorganized and expanded to clearly present evaluation metrics, data resources, extended experiments, limitations, and usage clarifications.

We hope that these revisions adequately address the reviewers' concerns and significantly strengthen the clarity and completeness of the paper.

We sincerely thank the reviewers for their time and valuable feedback.

---

### Meta-Review · Area_Chair_Sedb · 2026-01-06

**Summary:**

* *Limitation on technical novelty*: Some reviewers pointed out that the paper lacks conceptual and methodological novelty, as hierarchical MoE and hierarchical labels have been used in computer vision for years. While some reviewers considered the technical innovation to be 'fair', they did not deem it groundbreaking, as it primarily involves applied machine learning in the field of genomics.

* *Relying on the *ground truth* taxonomy*: The proposed method demands explicit taxonomic labels (Kingdom, Phylum, etc.) to route sequences during training. Reviewer NJUT noted that this makes the model heavily dependent on a *perfect* classification tree, which may not always be available for novel or poorly-defined species (e.g., in metagenomics)!

* *Poor presentation*: Most of the Reviewers mentioned that inconsistencies in the representation of Figures and Tables, typos, (and I can still see some of these issues in the current version.)

* *Additional ablation studies and fair comparison* Reviewer n5aV correctly raised two questions in this regard and asked for the fair comparison since GENE-M1’s success might just come from being trained on a better, more balanced dataset

**Reviewer Concerns:**

During the rebuttal phase, the authors have made a respectable effort to address empirical gaps via the new results in Appendix A.1–A.5; however, these additions do not mitigate the core issues of limited conceptual novelty and restricted generalizability. The framework’s strict reliance on ground-truth taxonomy and its iterative, rather than transformative, contribution to the MoE literature, coupled with ongoing presentation weaknesses, means the paper lacks the necessary rigor and breadth to warrant acceptance

**Reviewer Scores:**

* The Reviewer n5aV suggested 4 as a rating with Confidence of 4 and might increase because of the results in Appendix A3.

* The Reviewer weEp gave a rating of 6 with confidence. 3, and will maintain this rating.

* The Reviewer s2o1 rated the paper 4 with Confidence of 4, and he/she mentioned that he/she plans to keep the score 4.

* The Reviewer NJUT gave 4 as a rating with confidence of 4 and will maintain this rating.

---

### Decision · Program_Chairs · 2026-01-26

Reject